# Reconstitution of SPO11-dependent double-strand break formation

Zhi Zheng[1,2], Lyuqin Zheng[2,6], Meret Arter[2,6], Kaixian Liu[2,6], Shintaro Yamada[2,3], David Ontoso[2,4], Soonjoung Kim[2,5] & Scott Keeney[1,2,4✉]

Meiotic recombination starts with SPO11 generation of DNA double-strand breaks (DSBs)[1]. SPO11 is critical for meiosis in most species, but it generates dangerous DSBs with mutagenic[2] and gametocidal[3] potential. Cells must therefore utilize the beneficial functions of SPO11 while minimizing its risks[4]—how they do so remains poorly understood. Here we report reconstitution of DNA cleavage in vitro with purified recombinant mouse SPO11 bound to TOP6BL. SPO11–TOP6BL complexes are monomeric (1:1) in solution and bind tightly to DNA, but dimeric (2:2) assemblies cleave DNA to form covalent 5′ attachments that require SPO11 active-site residues, divalent metal ions and SPO11 dimerization. SPO11 can also reseal DNA that it has nicked. Structure modelling with AlphaFold 3 suggests that DNA is bent prior to cleavage[5]. In vitro cleavage displays a sequence bias that partially explains DSB site preferences in vivo. Cleavage is inefficient on complex DNA substrates, partly because SPO11 is readily trapped in DSB-incompetent (presumably monomeric) binding states that exchange slowly. However, cleavage is improved with substrates that favour dimer assembly or by artificially dimerizing SPO11. Our results inform a model in which intrinsically weak dimerization restrains SPO11 activity in vivo, making it exquisitely dependent on accessory proteins that focus and control DSB formation.

DSBs generated by SPO11 activity are a nearly universal feature of meiosis because they initiate the homologous recombination that supports chromosome pairing, chromosome segregation and genome diversification, which are fundamental to gamete formation and sexual reproduction[1,6,7]. In mice and humans, absence of SPO11 or its partner TOP6BL results in infertility owing to meiotic failure[8–11].

SPO11 is evolutionarily related to the DNA-cleaving subunit of archaeal topoisomerase VI (topo VI)[12,13]. The topo VI holoenzyme is a tetramer of two A subunits (Top6A, DNA cleaving) and two B subunits (Top6B, GHKL-type ATPase)[14,15] (Fig. 1a). Attack by the Top6A subunits on opposite strands of a duplex using active-site tyrosines breaks the DNA and produces covalent 5′-phosphotyrosyl links[16]. A winged-helix (WH) domain in Top6A bears this tyrosine (equivalent to Y138 in mouse SPO11[17]), and a separate Rossmann fold known as a Toprim domain coordinates metal ions that are also needed for catalysis[14,15,18]. DNA strand breakage requires that the tyrosine of one Top6A protomer interact with the $Mg^{2+}$-binding pocket of the second Top6A, forming a hybrid active site and necessitating Top6A dimerization for catalysis (Fig. 1a).

SPO11 covalently attaches itself to DNA during DSB formation in vivo[13,19], indicating that it too cuts DNA via a topoisomerase-like transesterase reaction. However, despite the nearly three decades since SPO11 was recognized as the DNA-cleaving initiator of meiotic recombination[12,13], key questions remain about its molecular mechanism.

Top6B homologues have been identified more recently[9,20]. The *Saccharomyces cerevisiae* homologue Rec102 has long been known to be important for Spo11 activity[21,22], but its relationship with Top6B had not been recognized. The yeast and fly Top6B homologues lack the GHKL domain that mediates ATP-dependent dimerization in topo VI, whereas the mouse and plant counterparts contain a degenerate version of this domain[9,20,23].

Previous attempts to purify recombinant proteins from various species have so far been unable to reconstitute SPO11 transesterase activity (Supplementary Discussion 1). We previously reported characterization of recombinant *S. cerevisiae* Spo11 'core complexes' (Spo11 plus the Top6B homologue Rec102 and the phylogenetically restricted interaction partners Ski8 and Rec104)[24,25]. Surprisingly, yeast core complexes are 'monomeric' in solution (1:1:1:1 stoichiometry) rather than having the expected 2:2:2:2 stoichiometry that is predicted from the stable dimerization of Top6A[14,15,18]. Monomeric core complexes bind with high affinity (sub-nanomolar dissociation constant ($K_d$)) to DNA ends that mimic the cleavage product in having 2-nucleotide (nt) 5′ overhangs[24,25]. To date, however, no DNA cleavage activity has been reported for these complexes.

Here we report the purification and characterization of recombinant mouse SPO11–TOP6BL complexes, reconstitution of DNA cleavage in vitro and structural models of dimeric complexes bound to DNA. Our findings suggest that a weak dimer interface is a hallmark of SPO11 that restrains its potentially dangerous ability to damage the genome and renders SPO11 dependent on other factors to control when and where it acts.

[1]Louis V. Gerstner Jr. Graduate School of Biomedical Sciences, Memorial Sloan Kettering Cancer Center, New York, NY, USA. [2]Molecular Biology Program, Memorial Sloan Kettering Cancer Center, New York, NY, USA. [3]The HAKUBI Center for Advanced Research and Department of Aging Science and Medicine, Graduate School of Medicine, Kyoto University, Kyoto, Japan. [4]Howard Hughes Medical Institute, Memorial Sloan Kettering Cancer Center, New York, NY, USA. [5]Department of Microbiology and Immunology, Institute for Immunology and Immunological Diseases, Yonsei University College of Medicine, Seoul, Republic of Korea. [6]These authors contributed equally: Lyuqin Zheng, Meret Arter, Kaixian Liu. ✉e-mail: s-keeney@ski.mskcc.org

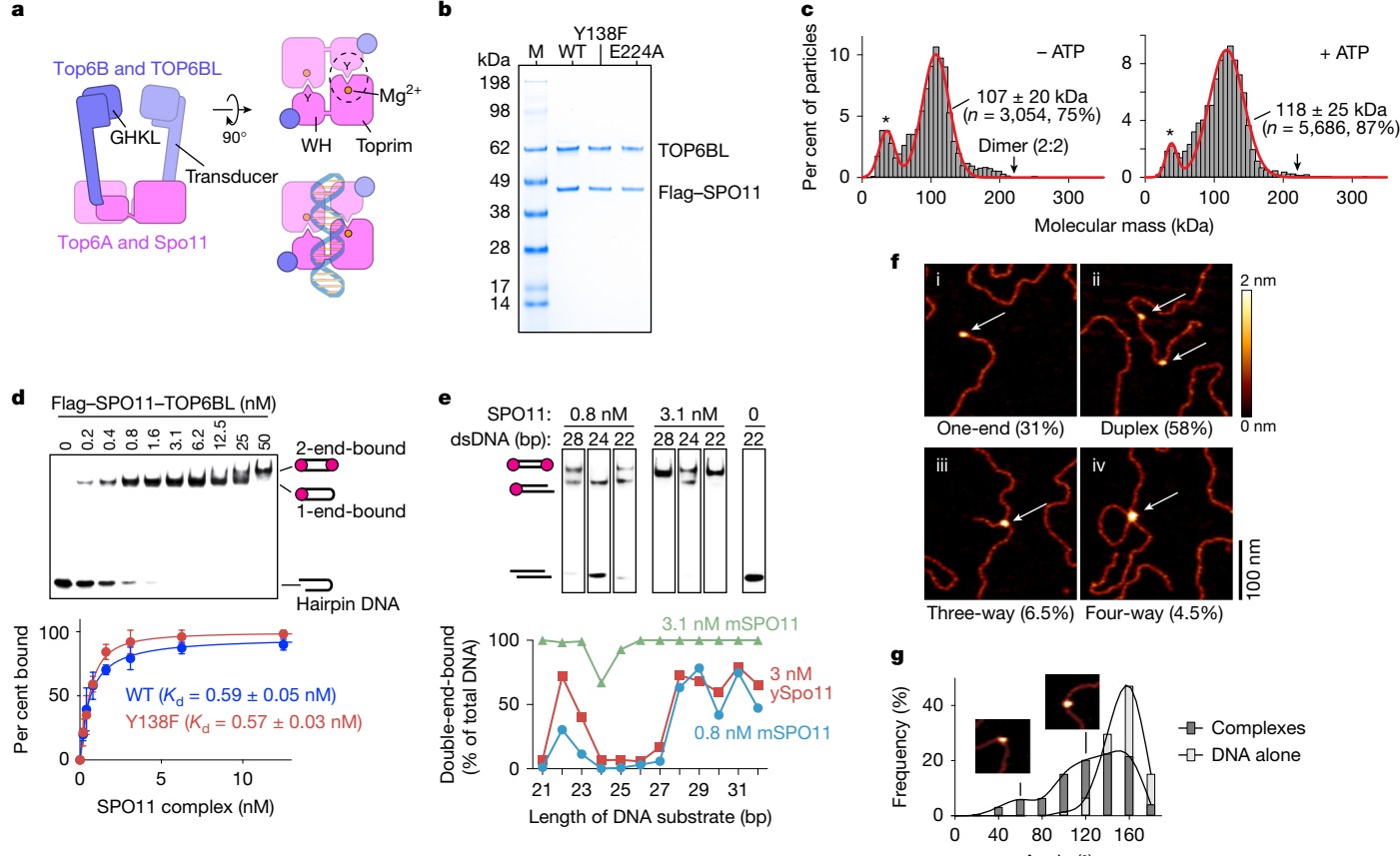

**Fig. 1 | Purification and DNA-binding activity of SPO11–TOP6BL complexes.**
**a**, Domain organization of dimeric SPO11 complexes and topo VI. Left, side view. Right, top views into the DNA-binding channel, without (top) and with (bottom) DNA. The catalytic tyrosine (Y), metal-binding pocket and hybrid active site (dashed circle) are shown. Figure adapted from ref. 24, CC BY 4.0 (https://creativecommons.org/licenses/by/4.0). **b**, Coomassie-stained SDS–PAGE of purified Flag–SPO11–TOP6BL preparations (0.5 μg each). Mutant protein purifications were performed at least twice and wild-type purification was performed more than five times with similar results. For gel source data, see Supplementary Fig. 1. M, molecular mass markers; WT, wild type. **c**, Mass photometry of monomeric SPO11–TOP6BL complexes (28 nM) with or without 5 mM ATP. Particle counts (grey bars), gaussian fits (red lines), fitted mean ± s.d. and percentages of total particles are shown. Asterisks indicate background, which is also present in blanks. **d**, Top, EMSA of SPO11 complexes binding to a

5′-labelled 25-bp hairpin substrate with a two-nucleotide 5′ overhang end. Bottom, quantification (mean ± s.d. of $n = 3$ experiments; apparent $K_d$ given as mean ± s.e.m.) is shown. **e**, DNA length dependence for double-end binding. Top, selected lanes from gel shift assays show binding of SPO11 complexes to DNA of the indicated lengths with two-nucleotide 5′ overhangs on both ends (see full gels in Extended Data Fig. 1d). Bottom, quantification of double-end binding, with yeast Spo11 (ySpo11) data[27] for comparison. mSPO11, mouse SPO11. **f**, AFM analysis of binding to linearized plasmid DNA. Examples are shown of binding to ends (one-end), internally on duplex DNA (duplex), junctions of three DNA arms (three-way) and junctions of four DNA arms (four-way). Percentages of binding particles are shown for $n = 200$ particles. **g**, Histogram of bending angles ($n = 217$ particles bound internally on duplex DNA) compared with randomly chosen protein-free positions ($n = 138$). Examples are from subpopulations with modal values of approximately 60° and 120°, similar to yeast[25].

## Purification of SPO11–TOP6BL complexes

We purified complexes of Flag-tagged SPO11 with TOP6BL after expression in cultured human cells (Fig. 1b and Extended Data Fig. 1a,b). The proteins co-eluted following size-exclusion chromatography (SEC) consistent with a 1:1 complex (110.3 kDa) (Extended Data Fig. 1b). This monomeric configuration (we use 'monomeric' and 'dimeric' to refer to the number of SPO11 protomers in a complex) was confirmed by mass photometry, in which most particles approximated the predicted size for a 1:1 complex and few if any matched a 2:2 stoichiometry (Fig. 1c, left). The complexes remained monomeric in the presence of ATP (Fig. 1c, right), similar to purified TOP6BL alone[26]. Monomeric mouse SPO11 complexes thus resemble yeast Spo11 core complexes[24,25] and are unlike Top6A alone or in complex with Top6B[14,15,18].

## High-affinity binding to DNA ends

In electrophoretic mobility shift assays (EMSAs) with 25-bp hairpin substrates with a 2-nt 5′ overhang, 0.2 nM protein yielded a single

discrete shifted species (Fig. 1d). This high-affinity DNA binding (apparent $K_d \approx 0.6$ nM) was not diminished by Y138F mutation (Fig. 1d and Extended Data Fig. 1c). A second shifted species appeared at higher concentrations (≥25 nM), probably reflecting lower-affinity binding of a second protein complex to the hairpin end.

Similar to yeast core complexes[27], two mouse SPO11 complexes can bind separately to each end of a DNA if the two 5′ overhangs are separated by at least 28 bp (Fig. 1e and Extended Data Fig. 1d). At low protein concentration (0.8 nM), double-end binding was disfavoured for duplexes of 24–27 bp, but could occur on even shorter DNA (22–23 bp) (Fig. 1e). Analogous behaviour of the yeast core complex appears to reflect steric clashes that are relieved by relative rotation of the two end-bound complexes[24]. Unlike in yeast, however, increasing the mouse protein concentration largely or completely restored double-end binding to all of the short DNA substrates tested (Fig. 1e and Extended Data Fig. 1d). The mouse protein thus appears to be more permissive of binding to DNA in close quarters.

In atomic force microscopy (AFM) analysis, most protein particles bound internally to linear plasmid substrates, frequently at DNA bends

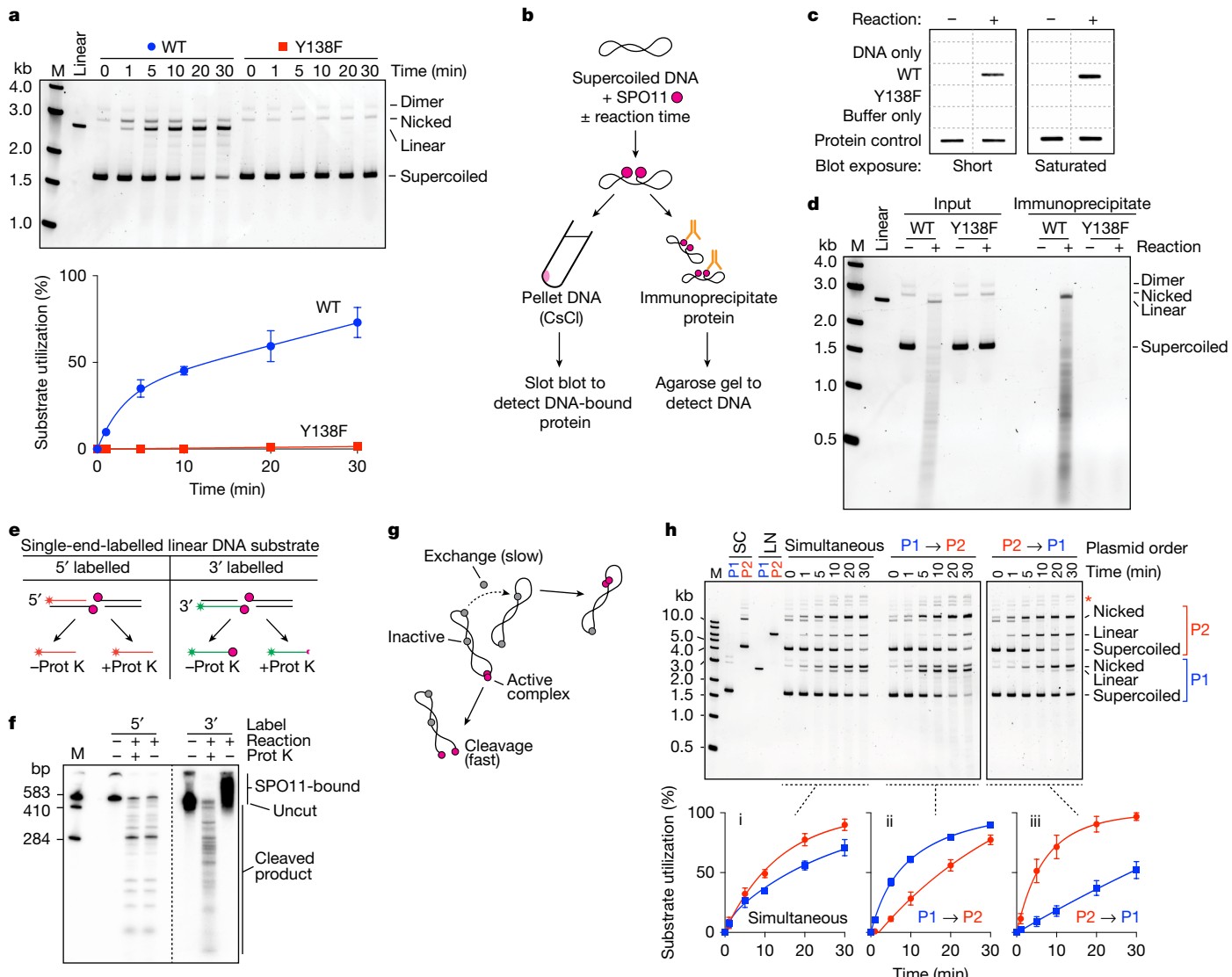

**Fig. 2 | Reconstitution of SPO11-dependent DSB formation in vitro. a**, DNA cleavage. Reactions contained 4 ng μl$^{-1}$ pUC19 DNA, 100 nM SPO11 complexes and 5 mM MnCl$_2$. Top, representative gel (deproteinized samples were separated on agarose gels and stained with SYBR Gold). Bottom, quantification (mean ± s.d. of $n$ = 3 experiments). **b**–**d**, Covalent attachment of SPO11 to cleaved DNA. **b**, Schematic summarizing experiments; details are in Methods. Mixtures of SPO11 complexes with DNA were quenched with SDS or incubated at 37 °C. **c**, Samples were then centrifuged through CsCl cushions and immuno-slot-blotted with anti-Flag antibodies (two exposures shown, performed once). **d**, Samples were immunoprecipitated with anti-Flag antibodies and separated on agarose gels (the DNA smear reflects multiple DSBs; performed twice with similar results). **e**, Schematic depicting covalent protein association with radiolabelled strands (asterisks). Prot K, proteinase K. **f**, SPO11 attachment to 5′ ends. A 583-bp fragment from pUC19 was 5′ or 3′ radiolabelled on one end, then incubated with SPO11 complexes and separated by denaturing PAGE with or without deproteination (performed twice with similar results). **g**, Model explaining the biphasic reaction kinetics. **h**, Substrate order of addition determines reaction rate. SPO11 complexes (100 nM) were incubated with 4 ng μl$^{-1}$ each of pUC19 (P1, 2.7 kb) and plasmid mp134 (P2, 6.8 kb). Protein was mixed with both plasmids on ice before initiating reactions by transfer to 37 °C (simultaneous), or was incubated with one plasmid on ice and the second plasmid was added immediately before transfer to 37 °C (P1→P2 and P2→P1). Timed samples were quenched with SDS and deproteinated before agarose gel electrophoresis. Supercoiled (SC) or linear (LN) plasmids were also run as size markers. Top, representative gels. Bottom, quantification (mean ± s.d. of $n$ = 3 experiments). Asterisk indicates likely P2 multimers.

(Fig. 1f(ii),g), but nearly a third were bound to DNA termini (Fig. 1f(i)). This finding confirms preferential end binding because internal sites are in molar excess over termini. A minor fraction of particles were located where duplex segments crossed or where three DNA arms emanated out (Fig. 1f(iii),(iv)). These patterns are congruent with yeast[25].

## Reconstituting DSB formation

To test for DNA cleavage, we incubated SPO11 complexes with super-coiled plasmid DNA and Mn$^{2+}$. We expected no activity without the essential cofactors[1] MEI4, REC114, IHO1 and MEI1. Remarkably, however,

wild-type complexes exhibited robust DNA cleavage that was not seen with the catalytic mutant SPO11(Y138F) (Fig. 2a). Both nicks and DSBs were generated, with nicks initially predominating: around 60% of broken plasmids at 1 min had sustained only a nick.

Transesterase activity should leave SPO11 covalently attached to the DNA. We tested this with reciprocal experiments (Fig. 2b). First, we purified the cleaved DNA by ultracentrifugation through a CsCl cush-ion to pellet DNA plus covalently attached protein while free protein floated on top. Flag–SPO11 in the pellet was detected by immunob-lotting. Wild-type SPO11 copurified with the DNA as expected, but no copurifying protein was detected in negative controls that used

SPO11(Y138F) or that allowed SPO11 to bind but not react with the DNA (Fig. 2c). Second, we immunoprecipitated SPO11 and then assayed for coprecipitated DNA. As expected, broken DNA was recovered in the immunoprecipitate from reactions with wild-type SPO11, but not from negative controls (Fig. 2d).

To test whether SPO11 attached specifically to 5′ strand termini[28–30], we digested linear DNA that was either 5′- or 3′-radiolabelled on one end. Radioactive cleavage products of the 5′-labelled substrate are not expected to be protein-bound, and so should run identically on denaturing polyacrylamide gel electrophoresis (PAGE) with or without protease digestion (Fig. 2e, left). Conversely, cleavage of the 3′-labelled substrate should leave SPO11 attached to the radioactive strand, necessitating protease digestion for the labelled products to run into the gel (Fig. 2e, right). These predictions were met (Fig. 2f); thus we have recapitulated the transesterase activity expected for SPO11.

DNA cleavage required divalent metal ions, with $Mn^{2+}$ supporting greater activity than $Mg^{2+}$, which yielded more nicks relative to DSBs (Extended Data Fig. 2a). $Ca^{2+}$ supported weak nicking but few DSBs (Extended Data Fig. 2a), in contrast to topo VI from *Saccharolobus shibatae*, for which $Ca^{2+}$ supports robust DSB formation[16,31]. DNA cleavage was greatest at physiological or higher temperature (Extended Data Fig. 2b) and showed a broad pH optimum from 7.5 to 8.5 (Extended Data Fig. 2c). Positively supercoiled and relaxed plasmids were cut effectively, but activity was higher on negatively supercoiled DNA (Extended Data Fig. 2d,e). Linear DNA was also cut effectively (Fig. 2f and Extended Data Fig. 2f). On the basis of this optimization, we used standardized reaction conditions of 1 mM $MnCl_2$, pH 7.5 and 37 °C for additional experiments unless indicated otherwise.

## Slow exchange limits DSB competence

Cleavage time courses typically displayed a fast initial rate with little or no lag, followed by a longer phase of slower cutting (for example, Fig. 2a and Extended Data Fig. 2d,e). The slower phase is not an artefact from underestimating total cleavage as substrates incur multiple nicks and DSBs, because it started at similar time points on different substrates irrespective of the fraction of DNA cleaved (for example, Extended Data Fig. 2d,e).

To account for these findings, we hypothesized that SPO11 complexes rapidly bind DNA when the mixtures are assembled on ice, but only a subset of the protein-bound complexes are in a cleavage-competent state (possibly dimers) that cut DNA quickly when shifted to 37 °C (Fig. 2g). We further posited that the sluggish second phase mostly reflects rate-limiting exchange of non-productive SPO11–DNA complexes (possibly monomers) to form new cleavage-proficient complexes, which is slow relative to cleavage at least in part because of the high affinity (low off rate) of DNA binding and the excess of available monomer binding sites.

To test this hypothesis, we incubated SPO11 complexes with plasmid DNA on ice, then challenged the mixture with free DNA of a different size. In control experiments, both plasmids were cut with two-phase kinetics when mixed with the protein simultaneously (Fig. 2h(i)). The larger plasmid was consumed more quickly as a percentage, as expected because it is a bigger target and thus fewer molecules were present. By contrast, in the staged reactions, the first plasmid was cut faster than either the plasmid added later or the same plasmid when in the simultaneous mixing regime (Fig. 2h(ii),(iii)). Moreover, cleavage of the first plasmid again showed two-phase kinetics, whereas cleavage of the second plasmid followed a monophasic time course that resembled the slow portion of the respective two-phase reaction. These findings support our interpretation (Fig. 2g).

## Structure of a pre-DSB dimeric complex

When queried with two SPO11–TOP6BL complexes, a DNA duplex and four $Mg^{2+}$ ions, AlphaFold 3 reproducibly predicted a dimeric protein assembly bound to bent DNA[5] (Fig. 3a–d, Extended Data Fig. 3a–h, Supplementary Fig. 2 and Supplementary File 1). Supporting the validity of the model, it is congruent with empirical structures and functional data for homologous proteins[14,15,18,24,32,33] (Supplementary Discussion 2). For example, the protein resembles crystal structures of topo VI (Extended Data Fig. 3f,g). The DNA is positioned along a SPO11 channel cognate to the DNA-binding surface defined experimentally for yeast Spo11 and proposed for topo VI, and many of the protein–DNA contacts observed for yeast are recapitulated (Fig. 3a–d and Extended Data Fig. 4a–c). Further, the model predicts hybrid active sites with two $Mg^{2+}$ ions positioned within each Toprim domain coordinated by side chains of E224, D277 and D279, with the catalytic tyrosines nearby and close to opposite DNA strands (Fig. 3e). The predicted SPO11 dimer interface includes similar structural elements as Top6A dimers (Extended Data Fig. 5a,b). Finally, DNA bending was observed in mouse (Fig. 1g) and yeast[25] AFM experiments and in yeast cryo-electron microscopy (cryo-EM) structures[24], and was predicted for topo VI[33]. We conclude that the AlphaFold 3 models are plausible representations of dimeric pre-DSB complexes.

Several new insights emerged from these models. First, we considered whether AlphaFold 3 could explain why ATP did not support dimerization of SPO11–TOP6BL complexes (Fig. 1c), unlike topo VI[14,32]. When ATP was included, AlphaFold 3 predicted nucleotide-mediated GHKL domain dimerization matching a crystal structure[32] for *S. shibatae* Top6B, but not for TOP6BL (Extended Data Fig. 5c,d). The TOP6BL GHKL domain was predicted to have the expected secondary structure and overall fold (Extended Data Fig. 5e), but a structure-informed alignment indicated that TOP6BL lacks a conserved glycine-containing G1 box that contacts ATP[34] (Extended Data Fig. 5f).

Second, the predicted DNA bending accompanies unwinding of the two base pairs flanking the dyad axis (Fig. 3c). If correct, this deformation may position the scissile phosphates in the active sites and/or impart DNA strain to promote catalysis or disfavour religation. The bending and unwinding may also explain nonrandom base composition at favoured cleavage sites (addressed below). However, bend positions were variable, indicating that DNA sequence contributes little to the AlphaFold 3 prediction (Supplementary Discussion 2).

Third, the relative arrangement of the WH and Toprim domains does not match precisely with the yeast Spo11 cryo-EM structure that is thought to represent a post-DSB product-bound state[24]. However, the two structures can be aligned by a 13° rigid-body movement of the WH domain (Extended Data Fig. 5g), which is plausible given that the protein segment connecting the two domains is thought to be a flexible linker[18,25]. This finding supports the idea that DSB formation is accompanied by a SPO11 conformational change−relative motion of the WH and Toprim domains−that may influence cleavage reversibility[24].

Fourth, although protein–DNA contacts are similar to those for monomeric yeast core complexes (Supplementary Discussion 2), there are also new features. For example, an α-helix in the WH domain (residues 148–159) lies across the major groove at the bend (Fig. 3f). The side chains do not make obvious DNA contacts, but the position of the helix braces adjacent backbone-interaction patches: K133–R134–R165 plus the catalytic Y138 on one side, and R101–K106–K110 on the other. This helix might be important for stabilizing the bend and positioning Y138 near the scissile phosphate.

The predicted protein–DNA contacts are often rotationally asymmetric (Extended Data Fig. 3h and Supplementary Discussion 2). For the representative model shown, the same residues in each SPO11 monomer contact the DNA, but one monomer is shifted by approximately 1 bp relative to the other (Fig. 3d and Extended Data Fig. 4a,b). This places one catalytic tyrosine further from the scissile phosphate than the other (Fig. 3e), so this model may not be consistent with both active sites being simultaneously catalysis-competent. This asymmetry should be viewed cautiously because it varied between models and is not experimentally validated,

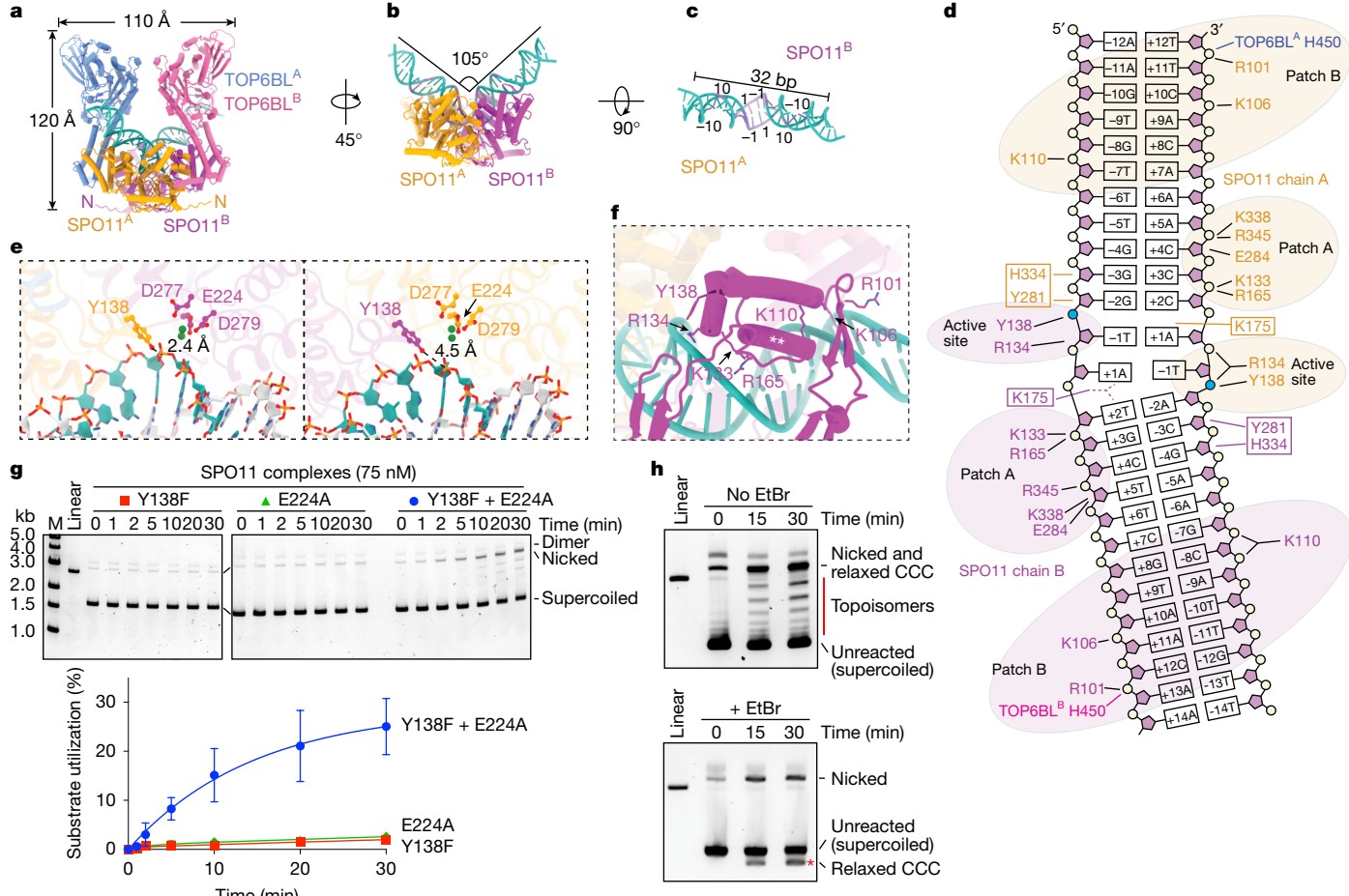

**Fig. 3 | Dimeric SPO11–TOP6BL complexes on DNA. a–c**, AlphaFold 3 model. **a**, Overview of the entire structure. **b**, Side view isolating SPO11 and DNA. **c**, Top view of the DNA. Positions relative to dyad axis are numbered. In **b,c**, DNA segments coloured darker blue showed biased base composition. A C-terminal α-helix and unstructured segment from TOP6BL[26] are omitted for clarity (see Extended Data Fig. 3a,b and Supplementary Discussion 2). **d**, Inferred contacts between the DNA (mostly sugar-phosphate backbone) and amino acid side chains. Groups of mostly positively charged residues (patches A and B) and active-site residues are shaded; boxed residues are predicted to occupy the minor groove. Bases are numbered relative to the presumed dyad axis. Scissile phosphates are shown in blue. **e**, Arrangement of catalytically important SPO11 residues in each of the two active sites (coloured as in **a**). Distances of each Y138 residue from its cognate scissile phosphate are shown. **f**, Detail view of the positioning helix (\*\*; residues 147–161) from a SPO11 WH domain, with adjacent DNA backbone contacts and the catalytic tyrosine indicated. **g**, Reconstituting DNA nicking with SPO11 mutants Y138F, E224A and a 1:1 mixture of Y138F and E224A. Top, representative agarose gels of cleavage reactions. Bottom, quantification (mean ± s.d. of $n$ = 3 experiments). **h**, Supercoil relaxation activity. Reactions containing mixtures of Y138F and E224A SPO11 complexes (as in **g**) were incubated for the indicated times then deproteinized and separated on agarose gels without (top) or with (bottom) 1 μg ml⁻¹ ethidium bromide (EtBr). The red line highlights the topoisomer ladder that disappears on EtBr-containing gels. The red asterisk highlights a new band on the EtBr gel that corresponds to plasmids that were relaxed covalently closed circles (CCC) after the reaction but that became positively supercoiled upon binding EtBr.

but it may explain why nicks are a prominent early product in cleavage reactions.

## Hybrid active sites in a SPO11 dimer

We sought direct evidence that strand cleavage is executed by hybrid active sites in a dimer by testing whether cleavage activity could be restored by cross-complementation of SPO11 proteins that were inactive on their own because of active-site mutations in either the WH domain (Y138F) or Toprim metal-binding pocket (E224A). We reasoned that mutant heterodimers would generate nicks because they would have one complete active site and one doubly mutated one.

The E224A mutant retained normal DNA end binding (Extended Data Fig. 5h) but was incapable of cutting DNA (Fig. 3g). By contrast, a mix of Y138F and E224A mutant proteins yielded nicking activity without appreciable DSBs (Fig. 3g). This result provides compelling evidence for formation of hybrid active sites.

## SPO11 relaxes and religates DNA

At later time points with mixtures of Y138F and E224A mutant SPO11, a ladder of bands appeared between supercoiled and nicked DNA (Fig. 3h). These bands disappeared when the gels contained ethidium bromide, coincident with appearance of a new band running faster than the unreacted substrate (asterisk in Fig. 3h, bottom). This behaviour is expected for partially relaxed, covalently closed topoisomers that become positively supercoiled upon binding ethidium bromide, and rules out that these might be linear species from double cutting at defined sites.

We conclude that SPO11 complexes can relax supercoils and reseal broken DNA strands by reversing the covalent 5′-phosphotyrosyl linkage. Topoisomer ladders were also generated by wild-type protein; they were most prominent when nicking occurred without substantial DSBs (for example, with Ca²⁺; Extended Data Figs. 2a and 5i), but were observed whenever nicked circles were present with both negatively

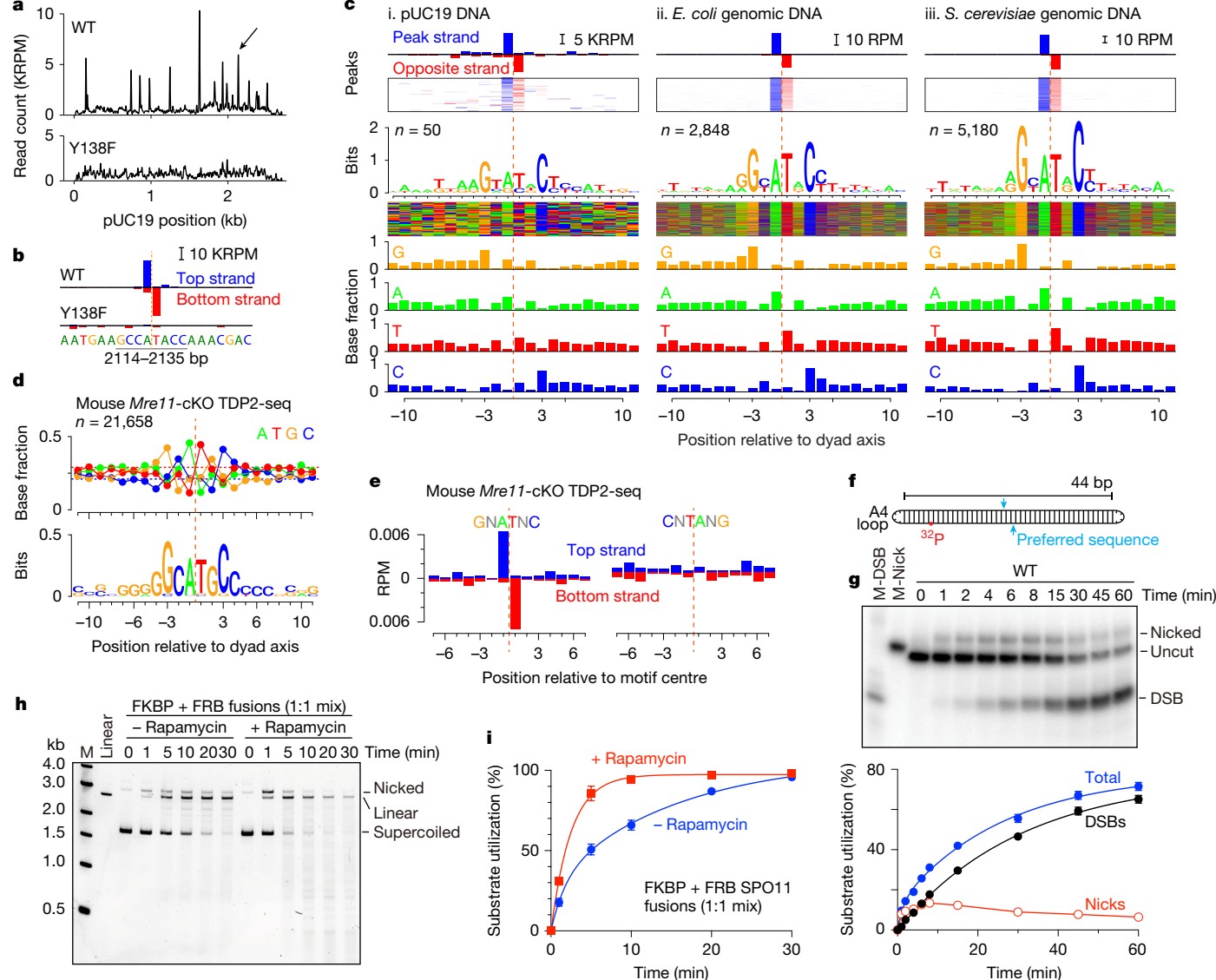

**Fig. 4 | SPO11 sequence preferences and stimulation of cleavage by dimerization. a,b,** Sequencing of pUC19 cleavage. Arrow in **a** indicates the position shown in **b** (orange dashed line, dyad axis). Read count (average of $n = 2$ replicates) is in thousands of reads per million mapped reads (KRPM). **c,** Sequence biases around cleavage sites on pUC19 (i) or genomic DNA from *E. coli* (ii) or yeast (iii). Read counts at peaks (top to bottom: averaged; as a heat map; sequence logo; sequence colour map (each horizontal line is one peak); and base fractions). RPM, reads per million mapped reads. **d,e,** Sequence bias in vivo (representative data from one of two experiments). TDP2-seq, tyrosyl DNA phosphodiesterase 2 sequencing. **d,** Fractional base composition (top) and sequence logo (bottom) are shown around preferred cleavage sites in hotspots[43] in *Mre11*-conditional

knockout (cKO) mice[35]. Note weaker bias compared with **c. e,** Sequencing reads were averaged across sites matching the in vitro bias (GNATNC, $n = 11,423$) or the non-preferred reverse sequence (CNTANG, $n = 9,018$). **f,g,** Efficient cleavage of an oligonucleotide substrate. **f,** Schematic of the substrate (details in Methods). **g,** Cleavage reactions contained 10 nM SPO11 complexes and 0.5 nM DNA. Top, representative autoradiograph with markers for nicks (M-Nick) and DSBs (M-DSB). Bottom, quantification (mean ± s.d. of $n = 3$ experiments). **h,i,** Stimulation of DNA cleavage by artificial SPO11 dimerization. Cleavage reactions contained 4 ng μl[−1] pUC19 DNA and 50 nM each of FKBP–SPO11–TOP6BL and FRB–SPO11–TOP6BL complexes with or without 5 μM rapamycin. **h,** Representative gel. **i,** Quantification (mean ± s.d. of $n = 3$ experiments).

and positively supercoiled DNA (Extended Data Fig. 2d). Thus, relaxation and resealing can occur on nicked DNA, but whether religation can also occur after a DSB is not clear. Implications and a potential mechanism are provided in Supplementary Discussion 3 and Extended Data Fig. 5j.

## SPO11 sequence bias

Preferred cleavage sites were apparent (Fig. 2f), so we mapped DSB positions by deep sequencing (Extended Data Fig. 6a). With wild-type protein and pUC19, sequencing reads were distributed across the plasmid but with prominent peaks that were not observed with SPO11(Y138F)

(Fig. 4a). Replicate maps were reproducible for wild-type protein, but not for the random background with SPO11(Y138F) (Extended Data Fig. 6b). SPO11 DSBs should have a 2-nt 5′ overhang, the middle of which is a two-fold rotational symmetry axis. Fill-in of the overhang should yield top- and bottom-strand reads that overlap by 1 bp, giving an offset of +1 in cross-correlation between strands (Extended Data Fig. 6a). This expectation was met for wild-type SPO11—but not SPO11(Y138F)—maps (Fig. 4b,c(i) and Extended Data Fig. 6c), thus the DSBs formed in vitro have the correct polarity.

The ten strongest cleavage sites in pUC19 showed bias for G at the −3 position relative to the dyad axis and C at the rotationally symmetric +3 position (Extended Data Fig. 6d). When the base composition was

averaged around 50 peaks in the sequencing maps, the biased base composition was extended to encompass the 8 bp surrounding the dyad axis (Fig. 4c(i)). The sequence signature was rotationally symmetric, as expected for a pair of SPO11 complexes engaging a pair of DNA half-sites. We emphasize that this is a base composition bias, not a strict sequence motif. Rotational symmetry does not indicate that individual SPO11 cleavage sites have to be palindromic.

We also generated maps using *Escherichia coli* or *S. cerevisiae* genomic DNA as the substrate to provide a larger repertoire of potential target sequences. Prominent cleavage sites with correct strand polarity were again observed with wild-type SPO11 but not SPO11(Y138F) protein (Extended Data Fig. 6e,f), with similar base composition biases (Fig. 4c(ii),(iii)). Similar results obtained with both supercoiled plasmid and relaxed genomic DNA show that superhelical tension is not required for the bias. However, the sequence signature strength correlated with the amount and complexity of the substrate—the signature was stronger with *E. coli* DNA than with pUC19 and stronger still with yeast DNA (Fig. 4c).

We attribute the stronger bias to two factors. First, reactions with yeast or *E. coli* DNA had threefold higher protein concentration but around ninefold higher DNA concentration than with pUC19. The increased DNA:protein ratio should attenuate dimerization by favouring more dispersed binding of monomeric protein complexes. Second, each DNA position competes with every other position for cleavage by SPO11. The more complex the substrate, the lower the concentration of any given site and the higher the concentration of competing sites. We suggest that these factors together affect the stringency of site selection, making SPO11 more dependent on optimal target–site matching with more complex substrates.

To test whether the same sequence signature applies in vivo, we sequenced DNA from *Mre11*-deficient mouse spermatocytes, which accumulate unresected DSBs[35]. Averaging over cleavage positions showed a weaker but similar base composition from −10 to +10 as in vitro, including enrichment for G and C at −3 and +3, respectively (Fig. 4d). Moreover, averaging over sites that match a favoured sequence (GNATNC) enriched for top- and bottom-strand reads at the correct positions, which was not seen with an unfavourable sequence (CNTANG) (Fig. 4e). We conclude that the intrinsic substrate preferences of SPO11 in vitro also shape the DSB landscape in vivo, but to a weaker extent, presumably because additional factors also contribute (Supplementary Discussion 4).

Unsurprisingly, the sequence bias spans the predicted footprint of dimeric SPO11 complexes on DNA (Fig. 3c,d). The bias for A and T at −1 and +1 corresponds to where the DNA is most distorted in the AlphaFold 3 model, and SPO11 is predicted to contact bases near the dyad axis and make multiple DNA backbone contacts that span the biased central region. Notable interactions in light of the sequence signature include Y281 and H334 contacting one strand between −2 and −4, patch A contacting the opposite strand nearby, and patch B contacting both strands near the −10 and +10 positions (Fig. 3d and Extended Data Fig. 4a).

The sequence may influence initial SPO11 binding, DNA bending and/or catalysis, but is unlikely to reflect direct readout of the bases because nearly all of the protein contacts are with the DNA backbone. Instead, the preferred base composition may favour an intrinsic DNA shape that supports SPO11 activity and/or appropriate bendability. In this vein, the preferred base composition predicts systematic variation in minor groove width (Extended Data Fig. 6g).

Even though the mouse and yeast proteins have similarly high affinity for DNA ends, have similarly sized footprints on DNA and share many of their DNA-binding residues, their sequence preferences differ markedly. Deduced from in vivo patterns[27,36–38], yeast Spo11 prefers A at −4 and −5 and C at −2, but displays little preference at the −3 position or the dinucleotide centred on the dyad axis (Extended Data Fig. 6h). It thus appears that fine-scale site selection by SPO11 is evolutionarily plastic.

## Fostering dimerization improves cleavage

Armed with knowledge of site preferences, we revisited the hypothesis that inefficient dimer formation limits overall cleavage activity. We reasoned that a DNA substrate optimized for SPO11 dimer assembly should improve cleavage efficiency. To test this, we made an oligonucleotide substrate containing a preferred cleavage sequence from pUC19 in the middle (Fig. 4b,f). Both DNA ends were $A_4$ hairpins to reduce SPO11 end-binding affinity, and the duplex arm lengths (22 bp each) were chosen to encourage only a single dimeric complex to bind at or near the middle. This substrate was bound with high affinity (apparent $K_d$ of 1.8 nM) and doubly bound DNA complexes were efficiently formed (Extended Data Fig. 7a).

As predicted, this substrate was cut well when most of the DNA molecules were bound by two SPO11 complexes (Fig. 4g). Both nicks and DSBs were formed (Fig. 4g and Supplementary Discussion 5), the reaction required Y138 of SPO11 (Extended Data Fig. 7b), and $Mn^{2+}$ was preferred over $Mg^{2+}$ (compare Fig. 4g with Extended Data Fig. 7c). A mixture of Y138F and E224A mutant proteins generated only nicks with covalently attached protein, as expected (Extended Data Fig. 7d,e). The reaction efficiency suggests that most of the protein preparation is active if it is able to assemble a dimer (Supplementary Discussion 6).

Finally, we tested whether cleavage efficiency could be increased by tethering two SPO11–TOP6BL complexes together. Leveraging the predicted arrangement of SPO11 N termini close to one another away from the DNA-binding surface (Fig. 3a), we fused SPO11 to the C terminus of either FKBP or FRB, which dimerize in the presence of rapamycin[39], and purified them with TOP6BL (Extended Data Fig. 7f). Equimolar mixtures formed rapamycin-dependent particles consistent with dimeric complexes (Extended Data Fig. 7g). As predicted, rapamycin greatly enhanced DNA cleavage by these mixtures, with a faster initial rate, more substrate consumed in the initial burst, and more multiply cut DNA at later time points (Fig. 4h,i). This is the expected result if a larger fraction of DNA-bound SPO11 started out in an active dimeric form in the presence of rapamycin.

## Discussion

Our results agree well with the accompanying papers from the Claeys Bouuaert[40] and Tong[41] laboratories. We consider the weak dimerization of SPO11 complexes to be a defining distinction from topo VI, attributable to a low intrinsic affinity for SPO11 self-association compared with Top6A and loss of ATP-driven dimerization by the GHKL domain. We suggest that weak dimerization is key to understanding SPO11 mechanism and regulation.

Conserved yeast Rec114, Mei4 and Mer2 form nucleoprotein condensates that recruit Spo11 core complexes[42]. This is proposed to establish chromosome axis-associated clusters of co-oriented SPO11 proteins that can bind and cut DNA on chromatin loops[25,27,42] (Extended Data Fig. 8a and Supplementary Discussion 7). Clustering results in high DSB formation efficiency per cluster despite a low efficiency per SPO11.

Our findings suggest the following additions. In vivo, where the amount and complexity of DNA target is high relative to available SPO11, weak dimer formation plus high-affinity DNA binding establish a kinetic trap that favours dispersed monomer binding to chromatin and disfavours catalytically competent dimers. SPO11 thus becomes dependent on accessory factors that increase the local SPO11 concentration through clustering and that prepay part of the entropic cost of dimerization by co-orienting SPO11 complexes. SPO11 is thus nearly exclusively active within higher-order chromosomal structures that can also regulate the timing, number, location and spacing of DSBs. This arrangement provides an elegant solution to the challenge of how to ensure that SPO11 carries out its essential function while minimizing the risks that poorly controlled DSBs would pose.

The mouse protein so far appears to be more permissive for DSB formation in vitro than yeast Spo11. AlphaFold 3 models suggest that mouse SPO11 may adopt similar conformations in monomers and dimers (Extended Data Fig. 8b,c), whereas yeast Spo11 may tend to adopt distinct conformations depending on the oligomeric state (Extended Data Fig. 8d). We speculate that the protein from mouse (and perhaps other species; Extended Data Fig. 8e) may be intrinsically more dimer-compatible. If so, SPO11 dimerization potential may itself be an evolutionarily variable, and thus tunable, property.

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

## Methods

### Mouse experiments

Mouse experiments were performed in accordance with US Office of Laboratory Animal Welfare regulations and were approved by the Memorial Sloan Kettering Cancer Center Institutional Animal Care and Use Committee (IACUC protocol number 01-03-007). Mice were housed in solid-bottom, polysulfone, individually ventilated cages (IVCs) (Thoren Caging Systems) on autoclaved aspen-chip bedding (PWI Industries); γ-irradiated feed (LabDiet 50531, PMI) and acidified reverse osmosis water (pH 2.5 to 2.8) provided ad libitum. The cages also contained Nestlets, EnviroDri and/or EnviroPaks as environmental enrichment. The IVC system was ventilated at approximately 30 air changes hourly. HEPA-filtered room air was supplied to each cage and the rack effluent was exhausted directly into the building's exhaust system. Cages were changed weekly in either a HEPA-filtered vertical flow change station or a class 2 type A biological safety cabinet. The animal holding room was maintained at 21.5 ± 1 °C, relative humidity between 30% and 70%, and a 12:12 h light:dark photoperiod. Mice were euthanized by $CO_2$ asphyxiation for tissue collection. Male germline-specific *Mre11*-cKO mice were generated by crossing *Mre11-flox* (ref. 44) and *Ngn3-cre* (ref. 45) alleles as described elsewhere[35]. No randomization or blinding was performed.

### Recombinant protein expression and purification

The full-length optimized coding DNA for mouse SPO11 (Uniprot: Q9WTK8-1) with a Flag tag followed by a TEV cleavage site at the amino terminus, and untagged TOP6BL (Uniprot: J3QMY9-1) was cloned into the pCDNA3.1(+) vector (Invitrogen). The SPO11 construct encodes for the β splicing isoform, which includes exon 2 that is skipped in the α isoform. SPO11β is both necessary and sufficient for most DSB formation in vivo, and the exon 2-encoded sequence is required for robust interaction with TOP6BL[9,19,46–48]. Although we cannot exclude that the Flag tag affects function, we note that even larger tags (FKBP or FRB) do not appear to interfere with activity and that the N-terminus of SPO11 is predicted to be unstructured in the AlphaFold 3 models. Point mutants (Y138F and E224A) were generated using QuikChange mutagenesis. Codon-optimized sequences encoding human FKBP1A (Uniprot: P62942), the FRB domain of human MTOR (Uniprot: P42345), and a glycine-serine (GS) linker were synthesized as gBlocks. These sequences were then cloned into the same vector as mouse SPO11, positioned between the Flag tag and the TEV cleavage site. Amino acid sequences of Flag-tagged wild-type SPO11 and the FKBP and FRB fusions are provided in Supplementary File 2.

FreeStyle 293-F cells (Invitrogen) were cultured in FreeStyle 293 expression medium (Gibco) under 5% $CO_2$ in a Multitron-Pro shaker (Infors, 120 rpm) at 37 °C. The cell line was not authenticated or tested for mycoplasma. When the cell density reached $1.2 \times 10^6$ cells per ml, the plasmids were co-transfected as follows. For a 1 l cell culture, 1 mg of plasmids were pre-incubated with 2 mg of 40-kDa linear polyethylenimine (PEI) (Polysciences) in 25 ml of Opti-MEM (Gibco) for 25–30 min. Transfection was initiated by adding the mixture to the cell culture. About 16 h after transfection, sodium butyrate was added to a final concentration of 10 mM. Transfected cells were cultured for an additional 32 h (for a total of 48 h) before collection.

For each batch of protein purification, 1 l of transfected cells was collected by centrifugation at 3,100g and resuspended in lysis buffer containing 25 mM HEPES-NaOH, pH 7.5, and 500 mM NaCl. The suspension was supplemented with a protease inhibitor cocktail, including 1 mM phenylmethylsulfonyl fluoride (PMSF) and 1× Halt Protease Inhibitor Cocktail (Thermo Scientific). Cells were lysed by sonication and centrifuged at 46,000g for 60 min. The supernatant was applied to anti-Flag M2 affinity gel (Sigma, A2220) and allowed to flow through by gravity at 4 °C. The resin was rinsed three times with lysis buffer. Proteins were eluted with lysis buffer containing 200 µg ml$^{-1}$ Flag

peptide (Sigma). The eluent was then concentrated using a 10-kDa cut-off Centricon (Millipore) and further purified by SEC (Superdex 200 Increase 10/300 GL, GE Healthcare) using UNICORN 7.6 (build 7.6.0.1306). Peak fractions were pooled and concentrated to 40 µl at a concentration of approximately 1.5 mg ml$^{-1}$. Aliquots were frozen in liquid nitrogen and stored at −80 °C. Protein concentration was determined by absorbance at 280 nm using NanoDrop 8000 (version 2.3.3).

### Oligonucleotide substrates for DNA binding and cleavage assays

Oligonucleotide sequences are provided in Supplementary Table 1. Hairpin substrates with a two-nucleotide 5′-overhang end for EMSAs were assembled by self-annealing oligonucleotides that were 5′-end labelled with [γ-$^{32}$P]ATP (Perkin Elmer) and T4 polynucleotide kinase (NEB), followed by purification by native PAGE. Substrates for double-end binding experiments were generated by annealing complementary oligos to create 2-nt 5′ TA overhangs at both ends. The oligos were mixed in equimolar concentrations (10 mM) in STE buffer (100 mM NaCl, 10 mM Tris-HCl pH 8, 1 mM EDTA), heated, and slowly cooled. The substrates were then 5′-end labelled with [γ-$^{32}$P]ATP (Perkin Elmer) and T4 polynucleotide kinase and purified by native PAGE.

The sequence of the oligonucleotide substrate for DNA cleavage was designed with one of the preferred cutting sites from pUC19 (Fig. 4a,b) placed in the middle of a 44 bp duplex flanked by A4 ssDNA loops (Fig. 4f). The 96-nt self-complementary oligonucleotide was first purified using a 15% polyacrylamide-urea gel (Invitrogen) and then labelled with [γ-$^{32}$P]ATP (Perkin Elmer) and T4 polynucleotide kinase. The purified oligonucleotide was then annealed by heating and slow cooling. T4 DNA ligase was then used to seal the DNA nick, and the circular ligated oligonucleotide was purified from linear unligated molecules using another 15% polyacrylamide-urea gel. The circular oligonucleotide was then reannealed by heating and slow cooling.

### Electrophoretic mobility shift assays

For hairpin substrates, binding reactions (10 µl) were carried out in 25 mM Tris-HCl pH 7.5, 7.5% glycerol, 100 mM NaCl, 2 mM dithiothreitol (DTT), 5 mM $MgCl_2$, and 1 mg ml$^{-1}$ bovine serum albumin (BSA) with 0.1 nM DNA and the indicated concentration of SPO11 complexes. Reactions were assembled on ice then incubated for 30 min at 30 °C and separated on a 6% DNA retardation gel (Invitrogen) at 100 V for 30 min. For binding to the oligonucleotide cleavage substrate, SPO11 complex was incubated with 0.1 nM DNA at 4 °C for 30 min in a buffer containing 25 mM HEPES-NaOH pH 7.5, 50 mM NaCl, 5 mM $MgCl_2$, 7.5% glycerol, 1 mg ml$^{-1}$ BSA, and 2 mM DTT. A 6% DNA retardation gel was pre-run at 100 V for 20 min with 0.5× Tris-borate supplemented with 5 mM $MgCl_2$ as the running buffer, then the binding reactions were loaded and separated at 100 V for 40 min. For both substrates, gels were dried, exposed to autoradiography plates, and visualized by phosphor imaging using Amersham typhoon control software (version 4.0.0.4). Quantification was performed using GelBandFitter (version 1.7)[49], and apparent $K_d$ values were calculated by nonlinear regression in GraphPad Prism 10 (version 10.3.0 (461) for Mac OS X).

### Atomic force microscopy

Linear plasmids for AFM were prepared by treatment of pUC19 with NdeI. SPO11–TOP6BL complexes were diluted to a final concentration of 4 nM in the presence of 1 nM (molecules) DNA in 25 mM HEPES-NaOH pH 6.8, 5 mM $MgCl_2$, 50 mM NaCl, 10% glycerol. Mixtures were incubated at 30 °C for 30 min. A volume of 40 µl of the DNA-protein mixture was deposited onto pretreated mica with 3-aminoproply-trietoxy silane (APTES) (Pelco Mica Disc, V1, Ted Pella) for 5 min. The sample-deposited mica was rinsed with 1 ml molecular biology grade deionized water and dried gently with a stream of nitrogen. AFM images were captured using an Nanowizard V (JPK Scanning Probe Microscope Control Program 8.0.59.1) in AC Mode Imaging at room temperature. AFM probe (RFESPA-75, Bruker) with nominal frequencies of approximately

75 kHz and nominal spring constant of 3 N m$^{-1}$ was used for imaging. Images were collected at a 2 Hz line rate with an image size of 2 × 2 μm at 512 × 512-pixel resolution. For data processing, images were exported with 3080-pixel resolution. The images were processed with JPK Data Processing software (version 8.0.59.1). ImageJ (version 1.54g) was used to quantify DNA bending angles.

## Mass photometry

Mass photometry experiments were conducted using a Refeyn TwoMP instrument. A ready-to-use 6-well sample cassette (Refeyn) was placed at the centre of a clean, ready-to-use sample carrier slide (Refeyn), with each well designated for a single measurement. To find the focus, 15 μl of fresh MP buffer was loaded into the well. For SPO11 complexes, the buffer contained 25 mM HEPES-NaOH (pH 7.5), 150 mM NaCl, and 2 mM DTT. For complexes containing SPO11 fused to FKBP and FRB, the buffer was prepared with or without 5 μM rapamycin. The focal position was identified and maintained throughout the measurement using an autofocus system based on total internal reflection. Purified SPO11 or an equimolar mixture (1:1) of FKBP–SPO11 and FRB–SPO11 complexes was diluted to a concentration of 240 nM in MP buffer. Subsequently, 1 μl of the diluted SPO11 or FKBP–SPO11 plus FRB–SPO11 complex was added to the buffer drop, resulting in a final protein concentration of 15 nM. After autofocus stabilization, movies were recorded for a duration of 60 s. Data acquisition was performed using Refeyn AcquireMP (version 2024.1.1.0) and data analysis was carried out with Refeyn DiscoverMP (version 2024.1.0.0). Contrast-to-mass calibration was performed using a BSA protein standard (Sigma), containing BSA monomers and dimers with molecular masses of 66.5 and 132 kDa. Statistical analysis was performed using DiscoverMP, where the distribution peaks were fitted with Gaussian functions to obtain the average molecular mass of each distribution component. Plotting was carried out using GraphPad Prism 10.

## DNA cleavage assays

Positively supercoiled pUC19 was prepared by incubating negatively supercoiled pUC19 (Thermo Scientific) with reverse gyrase (gift from S. Bahng of the K. Marians laboratory), followed by purification using a QIAquick PCR purification kit (QIAGEN) and verification via chloroquine-containing agarose gel electrophoresis as described[50]. Relaxed covalently closed pUC19 was generated by incubating negatively supercoiled pUC19 with *E. coli* topoisomerase I (NEB) and purified using a QIAquick PCR purification kit (QIAGEN). Linearized substrates were prepared by double digestion of pUC19 with EcoRI and SspI, followed by agarose gel extraction using a QIAquick Gel Extraction Kit (QIAGEN). For quantification, the linearized substrates were labelled at both 5′ ends using [γ-$^{32}$P]ATP (Perkin Elmer) and T4 polynucleotide kinase (NEB).

Typical plasmid reactions (70 μl) were carried out in a buffer containing 25 mM HEPES-NaOH pH 7.5, 1 mM DTT, 0.1 mg ml$^{-1}$ BSA (Sigma), 1 mM MnCl$_2$, and 4 ng μl$^{-1}$ pUC19 DNA. The reaction buffer was prepared on ice, and the recombinant proteins (usually 75–100 nM) were then added on ice. Unless otherwise stated, the reactions were incubated at 37 °C for the specified times. At each indicated time point, 11 μl of the reaction mixture was removed and terminated with 3 μl of STOP solution (375 mM EDTA, 5% SDS) and 1 μl of proteinase K (Thermo Scientific, 20 mg ml$^{-1}$), followed by incubation for 30 min at 50 °C.

The reaction products were then mixed with 6× DNA loading dye (NEB) and separated by electrophoresis on a 1.2% agarose gel (Lonza) in TAE buffer. Using a customized vertical agarose gel system (gel length 14.5 cm, CBS Scientific), separation was carried out for 90 min at 80 V. The gels were subsequently stained with SYBR Gold (Invitrogen) and imaged using a ChemiDoc MP imaging system (Bio-Rad Image Lab Touch Software (version 3.0.1.14)).

Gel images were quantified using Image Lab (version 6.1.0 build 7) and GelBandFitter. Cleavage activity is expressed as percent of substrate utilized—that is, the fraction of DNA molecules that had sustained at least one detectable nick or DSB. It is important to note that this fraction underestimates total strand cleavage when DNA molecules sustain multiple breaks. This is because we cannot distinguish whether a nicked molecule has been nicked once or multiple times, we cannot distinguish whether a molecule with a DSB has also been nicked, and we cannot reliably quantify the number of breaks within the smear that results from multiple DSBs (plus covalently closed topoisomers when present). For simplicity, substrate utilization was calculated as the percentage of nicked circles and full-length linear molecules (DSBs) relative to the total DNA in each lane, subtracting any nicked DNA present in control (no protein) reactions as background (method 1). When multiple DSBs were clearly present—that is, where clear smearing appeared on the gel at later time points—we quantified substrate utilization for those lanes instead as the disappearance of the supercoiled band compared to time zero (method 2). Details about which quantification method was used can be found in the source data files for each graph. For plotting purposes, substrate utilization time courses were fitted to two-phase association curves by nonlinear regression in Prism 10. These curves are only intended to aid in visualization of trends and should not be viewed as a theoretically valid way to estimate underlying rate parameters.

For oligonucleotide cleavage assays, the substrate shown schematically in Fig. 4f was generated by self-annealing and ligation of a 5′-$^{32}$P-labelled ssDNA oligonucleotide containing the preferred cleavage sequence (blue arrows) from pUC19 shown in Fig. 4b. Cleavage reactions were performed in a buffer containing 25 mM HEPES-NaOH pH 7.5, 5 mM MnCl$_2$ (or 5 mM MgCl$_2$), 0.1 mg ml$^{-1}$ BSA, 1 mM DTT, and 0.02% NP-40. Mixtures containing 10 nM SPO11 complexes and 0.5 nM radiolabelled oligonucleotide were assembled at 4 °C and then immediately transferred to 37 °C to initiate the reaction. At each specified time point, 5 μl of the reaction mixture was taken out and rapidly quenched with 1.2 μl of 375 mM EDTA. The mixture was then digested with proteinase K for 30 min at 50 °C. Samples were loaded onto a 15% polyacrylamide-urea gel in 1× TBE running buffer and electrophoresed at 180 V for 50 min. Gels were then dried, exposed to autoradiography plates, visualized by phosphor imaging, and analysed using GelBandFitter. For experiments mixing SPO11(Y138F) and SPO11(E224A), 5 nM of each mutant complex was used. For SPO11(Y138F) alone, 10 nM of the mutant complex were used.

Note that the nicked product in Fig. 4g runs more slowly than the marker (M-Nick) because of the residual amino acid(s) left after proteolysis. We would therefore expect the DSB product to also migrate more slowly than the marker (M-DSB) for the same reason, but it instead comigrated closely with the marker. It may be that proteinase K is more effective at removing SPO11 residues from a DSB end than from a nick, but we cannot exclude the alternative possibility that the cleavage is happening somewhere other than the preferred sequence that is cut when in intact pUC19.

## Assays for covalent protein–DNA complexes

For the CsCl ultracentrifugation and immunoprecipitation assays, cleavage reactions (60 μl) were carried out in a buffer containing 25 mM HEPES-NaOH pH 7.5, 1 mM DTT, 0.1 mg ml$^{-1}$ BSA, 5 mM MnCl$_2$, and 4 ng μl$^{-1}$ pUC19 DNA. The reaction buffer was prepared on ice, and the recombinant proteins (325 nM final concentration) were then added on ice. The reactions were either terminated immediately or incubated at 37 °C for 11 min. To terminate the reaction for the ultracentrifugation assay, 50 μl of the reaction mixture was combined with 83.3 μl of a stop solution to yield final concentrations of 30 mM EDTA, 0.5% Sarkosyl, 5 M guanidine HCl. To terminate the reaction for the immunoprecipitation assay, 50 μl of the reaction mixture was combined with 12.5 μl of a stop solution to yield final concentrations of 37.5 mM EDTA, 0.5% SDS.

An adaptation of the ICE (in vivo complex of enzymes) assay was used for the immunodetection of proteins covalently bound to DNA[51]. In a new 5 ml centrifuge tube, 2 ml of 150% (w/v) CsCl solution was added,

then 2 ml of buffer containing 10 mM Tris-HCl pH 8.0, 0.1 mM EDTA, and 0.5% Sarkosyl was layered on top. The stopped reaction mixture (133.3 µl) was then layered on top, and buffer was added until the tubes were full. The tubes were sealed, placed in a TN-1865 ultracentrifuge rotor (Thermo Scientific), and centrifuged at 42,000 rpm (-157,000$g$) for 17.5 h at 24 °C in a Sorvall wX+ Ultra Series ultracentrifuge (Thermo Scientific). The resulting DNA pellets with covalently bound proteins were washed with 70% ethanol and dissolved in 1× TE buffer (10 mM Tris-HCl pH 8.0, 0.1 mM EDTA) for 2 h at room temperature. Each sample was mixed with one-third volume of 25 mM sodium phosphate pH 6.5 buffer, then applied to a 0.45-µm nitrocellulose membrane (Bio-Rad) using a slot-blot vacuum manifold (Bio-Rad). No-protein negative controls (DNA only and buffer only) were included, and recombinant SPO11 complex (-10 ng total protein) was applied to adjacent slots as a control for detection. SPO11 protein was immunodetected using an anti-Flag–HRP monoclonal antibody (mouse, 1:1,000, Sigma A8592), followed by incubation with ECL Prime western blotting detection reagent (Amersham) and detection using a ChemiDoc MP imaging system (Bio-Rad).

For immunoprecipitation, the stopped reaction mixture (prepared as in the preceding paragraph for the ICE assay) was mixed with 237.5 µl of binding buffer (25 mM HEPES-NaOH pH 7.5, 500 mM NaCl, 5 mM EDTA, 1% Triton X-100) and 50 µl of anti-Flag magnetic agarose (Pierce A36797), which had been pre-washed twice with binding buffer. The mixture was incubated at room temperature with gentle shaking at 1,000 rpm for 60 min. After incubation, the magnetic agarose was washed twice with wash buffer (25 mM HEPES-NaOH pH 7.5, 150 mM NaCl, 5 mM EDTA, 1% Triton X-100), then 30 µl of 2× Laemmli sample buffer (Bio-Rad) and 1 µl of proteinase K (Thermo Scientific, 20 mg ml$^{-1}$) were added, followed by incubation for 30 min at 50 °C. The products were purified using the QIAquick PCR purification kit (QIAGEN), mixed with 6× DNA loading dye (NEB), and separated by agarose gel electrophoresis as in the DNA cleavage assays. The gels were subsequently stained with SYBR Gold (Invitrogen) and imaged using a ChemiDoc MP imaging system (Bio-Rad).

To investigate 5′ or 3′ SPO11 attachment, 10 µg of pUC19 was cleaved with EcoRI and labelled at the 5′ ends using T4 polynucleotide kinase (NEB) with [γ-$^{32}$P]ATP or at the 3′ ends using Klenow (NEB) with [α-$^{32}$P]dATP and [α-$^{32}$P]TTP. Following labelling, the substrates were further digested with SspI to produce single-end labelled fragments, which were subsequently purified by agarose gel electrophoresis and extraction using QIAquick Gel Extraction Kit (QIAGEN). Cleavage reactions were performed at a substrate concentration of 2.3 nM (0.87 ng µl$^{-1}$) in the presence of 25 mM HEPES-NaOH pH 7.5, 1 mM DTT, 0.1 mg ml$^{-1}$ BSA, and 5 mM MnCl$_2$. The reaction buffer was prepared on ice, and recombinant proteins (325 nM) were added on ice. Reactions were incubated at 37 °C for 2 h. To terminate the reactions, 70 µl of the reaction mixture was combined with 17.5 µl of stop solution to achieve final concentrations of 37.5 mM EDTA and 0.5% SDS. A 43.75 µl aliquot (half) of the stopped reaction mixture was treated with 1 µl proteinase K and incubated at 50 °C for 1 h. Samples, with or without proteinase K treatment, were then mixed with an equal volume of 2× urea TBE loading buffer (Invitrogen) and heated at 85 °C for 5 min. Subsequently, 10 µl of each sample was loaded onto a prewarmed 6% TBE-urea gel (Invitrogen) at 60 °C and electrophoresed at 180 V for 30 min. Gels were dried, exposed to autoradiography plates, and visualized using phosphor imaging.

## TDP2-seq

In vitro cleavage assays for sequencing were conducted in 70 µl reaction buffer containing 25 mM HEPES-NaOH pH 7.5, 1 mM DTT, 0.1 mg ml$^{-1}$ BSA and 1 mM MnCl$_2$. For cleavage of pUC19, reaction buffer was mixed with 4 ng µl$^{-1}$ DNA (final concentration) on ice, then recombinant SPO11–TOP6BL complex (100 nM) was added and the mixture was incubated at 37 °C for 30 min followed by inactivation with 1 µl proteinase

K solution (Thermo Scientific, 20 mg ml$^{-1}$) and 30 min incubation at 50 °C. Bacterial and yeast genomic DNA was purified by extraction with phenol:chloroform:isoamyl alcohol (25:24:1; Thermo Fisher) and ethanol precipitation from *E. coli* DH5α cells (Takara) or exponentially growing *S. cerevisiae* cells of the S288C strain background. For genomic DNA cleavage, reaction buffer was mixed with 2.5 µg DNA on ice. Recombinant SPO11 (300 nM) was then added and the mixture was incubated at 37 °C for 60 min and inactivated as above.

Sequencing libraries were prepared using a modified version of the S1-sequencing protocol[52–55]. In brief, testes from 16-week old (replicate 1) or 5-week old (replicate 2) mice were dissociated, and cells were embedded in agarose plugs as described[55]. For mapping in vitro SPO11 cleavage sites, DNA digested by SPO11 in vitro was embedded in plugs together with wild type C57BL/6J mouse testis cells (0.5–1 million cells per plug), whose DNA acted as carrier during library preparation. Two plugs were prepared for each experiment. Following proteinase K and RNase A treatment, plugs were equilibrated in TDP2 buffer according to the TopoGEN protocol (50 mM Tris-HCl pH 8, 0.15 M NaCl, 10 mM MgCl$_2$, 0.5 mM DTT, 30 µg ml$^{-1}$ BSA, 2 mM ATP) and incubated with purified human TDP2 protein (490 pmol per plug, TopoGEN) at 37 °C for 30 min for removal of covalently linked SPO11 from DNA ends. Subsequent steps (end fill-in with T4 DNA polymerase (NEB), ligation to biotinylated adaptors, DNA purification, DNA shearing, ligation to a second adaptor at the sheared end, and PCR amplification) were all performed as described[55] with the following minor modifications: (1) to reduce the loss of very small DNA fragments from diffusion out of the plugs, the washing step after ligation of the first adaptor was reduced from overnight to 1 h at 4 °C for experiments with pUC19; (2) the NEBNext End Repair Module (NEB, E6050S) was used for repair of DNA ends after shearing. DNA libraries were sequenced on the Illumina NextSeq 1000 platform (paired-end, 50 bp) using Real Time Analysis (version 3.1 or version 4.1) at the Integrated Genomics Operation at Memorial Sloan Kettering Cancer Center. BCL files were converted to FASTQ files using DRAGEN suite (version 4.2.7). Sample size ($n = 2$) was chosen based on prior evaluation of reproducibility of sequencing maps[55]. No statistical methods were used to predetermine sample size.

Sequencing reads were mapped against the pUC19, *E. coli* (ASM584v2), *S. cerevisiae* (sacCer3) or mouse (mm10) reference sequence using modified versions of previously published custom pipelines[54]. In brief, reads were mapped using bowtie2 (version 2.5.3)[56] with parameters -N 1 -X 1000. Uniquely mapped reads were extracted and assigned to the nucleotide immediately next to the biotinylated adaptor. Statistical analyses were performed using R versions 4.2.3 and 4.3.2. Telomeres (500 bp at the ends of each chromosome) and ribosomal DNA (chr. XII:459,400–460,900) in the *S. cerevisiae* genome were masked before downstream analyses. Mitochondrial DNA and the 2µ plasmid were also excluded. Raw and processed TDP2-seq data were deposited at the Gene Expression Omnibus (GEO) (accession number GSE275291). Mapping statistics are found in Supplementary Table 2.

The pUC19 maps in Fig. 4a were smoothed with 21-bp Hann filter to simplify the display. For Fig. 4d, preferred cleavage sites were identified within previously defined hotspots[43] in maps from *Mre11*-cKO mice, in which DSBs remain unresected[35]. Note the different vertical scales for base composition and sequence logo in Fig. 4d compared to the in vitro data in Fig. 4c, indicating weaker bias for in vivo data. For Fig. 4e, sites matching the in vitro bias (GNATNC, n = 11,423) or the non-preferred reverse sequence (CNTANG, n = 9,018) were identified within meiotic DSB hotspots, and strand-specific sequencing maps from *Mre11*-cKO mice were averaged over each group of sites.

## Peak calling and base composition analyses

Top- and bottom-strand peaks were separately called as nucleotide positions with >3,000, >10 and >15 RPM of strand-specific read counts in pUC19, *E. coli*, and *S. cerevisiae* TDP2-seq data, respectively. Positions with less read counts than those located at −1 and +1 positions

on top and bottom strands, respectively, were not defined as peaks because those reads could be false-positively enriched due to incomplete fill-in of 5′ overhangs at DNA ends before adaptor ligation. Base compositions were calculated for the strand from which TDP2-seq read was sequenced—that is, bottom-strand peaks were analysed with reorientation so that the nucleotide immediately next to the biotinylated adaptor should be located at −1 position relative to dyad axis on top strand. Sequence logos were generated using ggseqlogo (version 0.2)[57], with base composition biases corrected for the genome average G+C content. For *S. cerevisiae* Spo11 in vivo data, the local G+C content around mapped DSB sites (47.8%) was used instead because it differed substantially from the genome average (38.5%). Motif matches in Fig. 4e were identified using dreg from the European Molecular Biology Open Software Suite (EMBOSS version 6.6.0)[58] with default parameters.

## AlphaFold 3

The pre-DSB SPO11 dimer models were generated using the AlphaFold 3 online service (https://golgi.sandbox.google.com)[5]. The input comprised two full-length mouse SPO11 (Uniprot: Q9WTK8-1) and TOP6BL (Uniprot: J3QMY9-1) proteins, two complementary DNA sequences (Supplementary Table 1), and four magnesium ions. Multiple DNA sequences were used to evaluate dependence of the model on the DNA composition. The sequences were selected to represent different SPO11 cleavage sites in yeast genomic DNA, and they varied in length from 36 to 44 bp. Each session produced five top-ranked models. We selected one representative model for figure preparation; this model's coordinates are provided in Supplementary File 1. Structural alignments, analysis, and figure generation were performed in Chimera (version 1.18)[59] and ChimeraX (version 1.8)[60].

## Reporting summary

Further information on research design is available in the Nature Portfolio Reporting Summary linked to this article.

## Data availability

Raw and processed TDP2-seq data are available at the Gene Expression Omnibus (GEO) under accession number GSE275291. The AlphaFold 3 model used to generate most of the figures is provided in.pdb format as Supplementary File 1. The mouse genome assembly mm10 (also known as GRCm38) is available at https://www.ncbi.nlm.nih.gov/datasets/genome/GCF_000001635.20/. Source data are provided with this paper.

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

**Acknowledgements** This article is subject to the Open Access to Publications policy of the Howard Hughes Medical Institute (HHMI). HHMI laboratory heads have previously granted a nonexclusive CC BY 4.0 license to the public and a sublicensable license to HHMI in their research articles. Pursuant to those licenses, the author-accepted manuscript of this article can be made freely available under a CC BY 4.0 license immediately upon publication. The authors thank C. Claeys Bouuaert and M. Tong for discussion and sharing of unpublished data; K. Marians, S. Bahng, C. Lee, D. Remus and members of the Keeney and D. Patel laboratories for discussions and experimental advice; M. Lu for providing a *Mre11*-cKO mouse; the MSK Integrated Genomics Operation for sequencing; and Y. H. Kim for AFM imaging. MSK core facilities are supported by National Cancer Institute cancer centre support grant P30 CA08748. Z.Z. is supported in part by a Bruce Charles Forbes Predoctoral Fellowship. K.L. was supported in part by the Damon Runyon Cancer Research Foundation (DRG-[2389-20] to K.L.). M.A. was supported in part by an EMBO Long Term Fellowship (ALTF 905-2019 to M.A.). This work was supported by NIH grant R01 HD110120 (to S. Keeney and D. Patel); an MSK Basic Research Innovation Award (BRIA, to S. Keeney and D. Patel); and NIH grant R35 GM118092 (to S. Keeney). S. Keeney is an HHMI investigator. The funders had no role in study design, data collection and analysis, decision to publish or preparation of the manuscript.

**Author contributions** S. Keeney conceived the project. Z.Z., L.Z., M.A., K.L., D.O. and S. Keeney designed experiments. Specific experiments were performed as follows: Fig. 2c, D.O. and Z.Z.; AlphaFold 3 models and structure analysis, L.Z. and K.L.; DNA sequencing of in vitro cleavage products, M.A. and Z.Z.; sequencing of DSBs in vivo, M.A.; analysis of deep sequencing data, M.A. and S.Y.; oligonucleotide cleavage experiments (Fig. 4f,g and Extended Data Fig. 7a–e), L.Z. All other experiments were performed by Z.Z. S. Keeney contributed to analysis of all data. S. Kim contributed unpublished reagents and data for the *Mre11*-cKO mice. S. Keeney supervised the research. Z.Z., K.L., M.A. and S. Keeney secured funding. S. Keeney prepared figures and wrote the paper. All authors edited the manuscript.

**Competing interests** The authors declare no competing interests.

**Additional information**
**Correspondence and requests for materials** should be addressed to Scott Keeney.

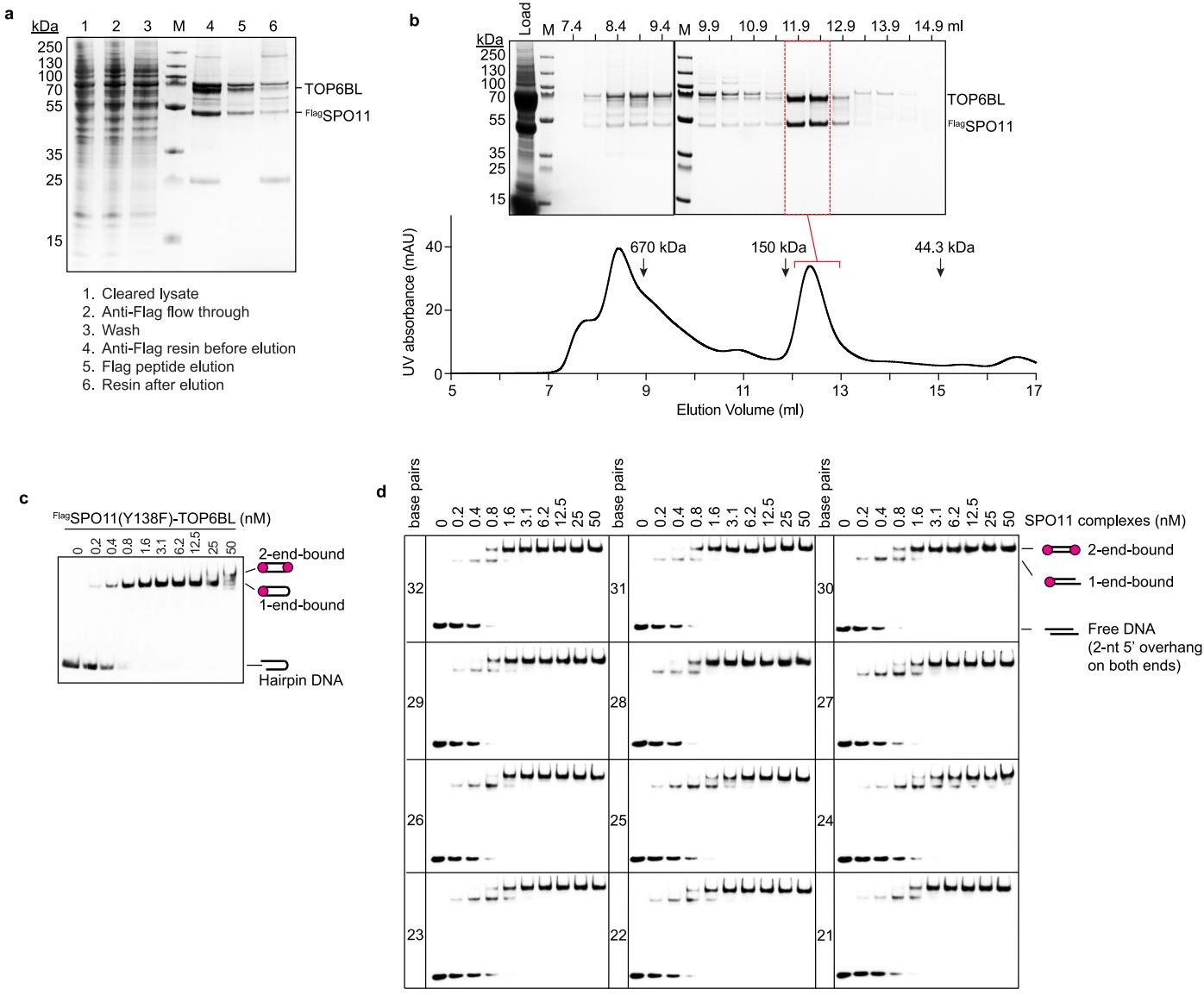

**Extended Data Fig. 1 | DNA binding by purified SPO11–TOP6BL complexes.**
**a**,**b**, Coomassie-stained SDS-PAGE gel of samples at the indicated steps from
a representative purification (**a**) and representative SEC profile (**b**). In **b**,
a Coomassie-stained SDS-PAGE gel of the indicated 0.5 ml fractions is above and
UV profile (mAU, milli-absorbance units at 280 nm) is below; elution positions
of size standards analyzed in a separate calibration run are indicated (arrows);
pooled fractions are indicated in red. Purification of wild-type protein was
performed more than five times with similar results. **c**, Representative EMSA

of SPO11-Y138F mutant complexes binding to a 25-bp hairpin substrate with
a two-nucleotide 5′ overhang end. Quantification is in Fig. 1d. **d**, DNA length
dependence for double-end binding. Full EMSAs are shown for the experiment
presented in Fig. 1e. This experiment was performed once. The steric constraints
that influence the length dependence of double-end binding are thought to
explain the minimum spacing between adjacent DSBs in vivo when multiple
Spo11 complexes cut the same DNA molecule in yeast or mouse[24,27].

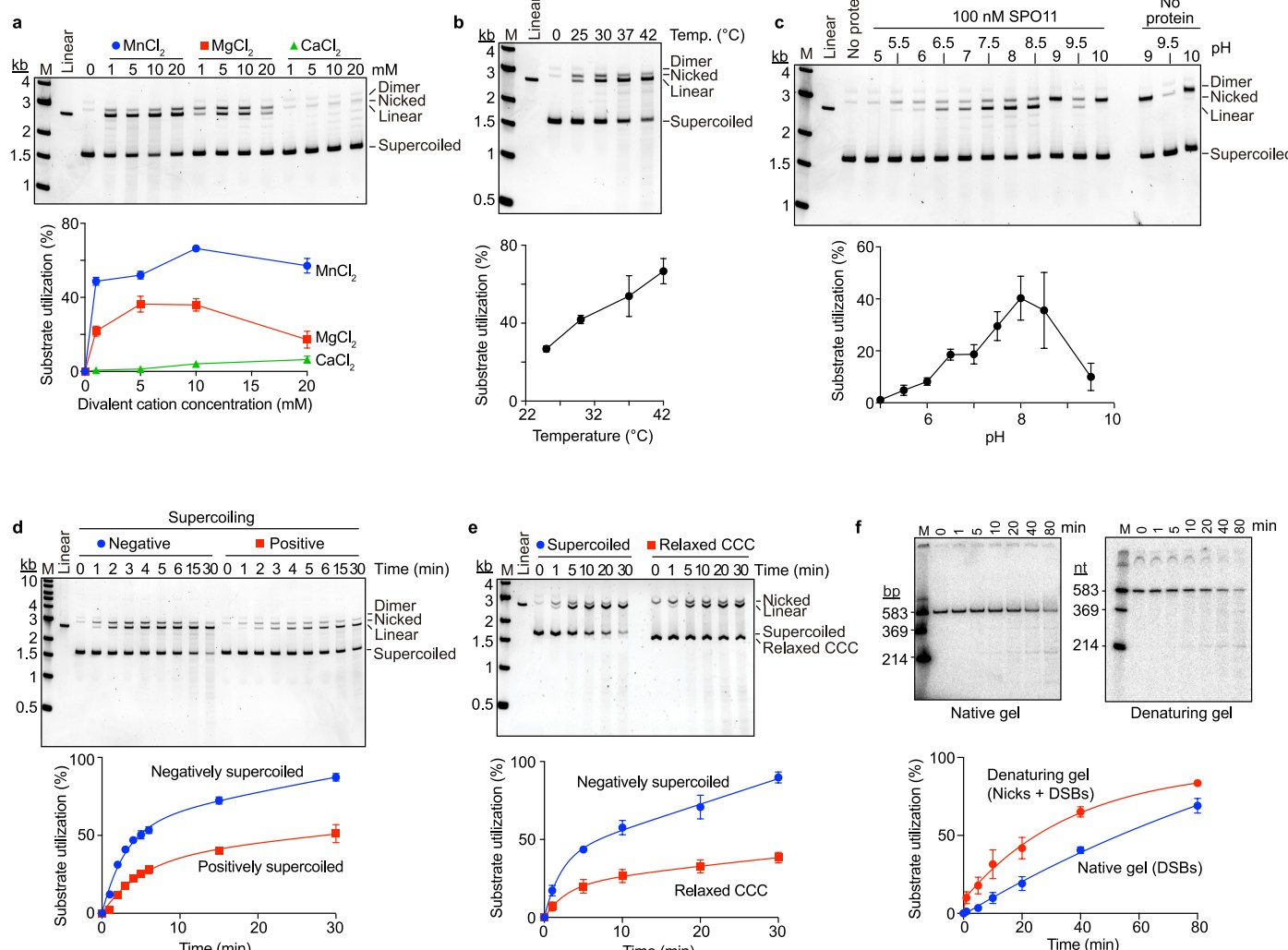

**Extended Data Fig. 2 | DNA cleavage optimization.** In all panels, representative gels are above, quantification is below (mean ± s.d. of n = 3 experiments). **a**, Metal ion dependence of DNA cleavage. SPO11–TOP6BL complexes (100 nM) were incubated with 4 ng/µl pUC19 DNA in the presence of the indicated concentration of MnCl$_2$, MgCl$_2$, or CaCl$_2$. **b**, Temperature dependence. Reactions contained 100 nM SPO11 complexes and 4 ng/µl pUC19 DNA with 1 mM MnCl$_2$. **c**, pH dependence. Reactions contained 100 nM SPO11 complexes and 4 ng/µl pUC19 DNA with 1 mM MnCl$_2$. The pH 9.0 and pH 10.0 conditions resulted in a high background of SPO11-independent nicking (right lanes), so these samples were omitted from the quantification. **d**, Comparison of positively and negatively supercoiled substrates. Reactions contained 100 nM SPO11 complexes and 4 ng/µl pUC19 DNA with 1 mM MnCl$_2$. **e**, Comparison of relaxed covalently closed circle (CCC) and negatively supercoiled substrates. Reactions contained 100 nM SPO11 complexes and 4 ng/µl pUC19 DNA with 1 mM MnCl$_2$. For this experiment, reaction products were separated on agarose gels containing ethidium bromide. **f**, Cleavage of a linear DNA substrate. Reactions contained 100 nM SPO11 complexes and 4 ng/µl of a linear DNA fragment from pUC19 (mix of cold and 5′ $^{32}$P-labeled on both ends) with 5 mM MnCl$_2$. Deproteinized reaction products were divided and aliquots were run separately on native PAGE to detect DSBs (left) and denaturing urea PAGE to detect both nicks and DSBs (right).

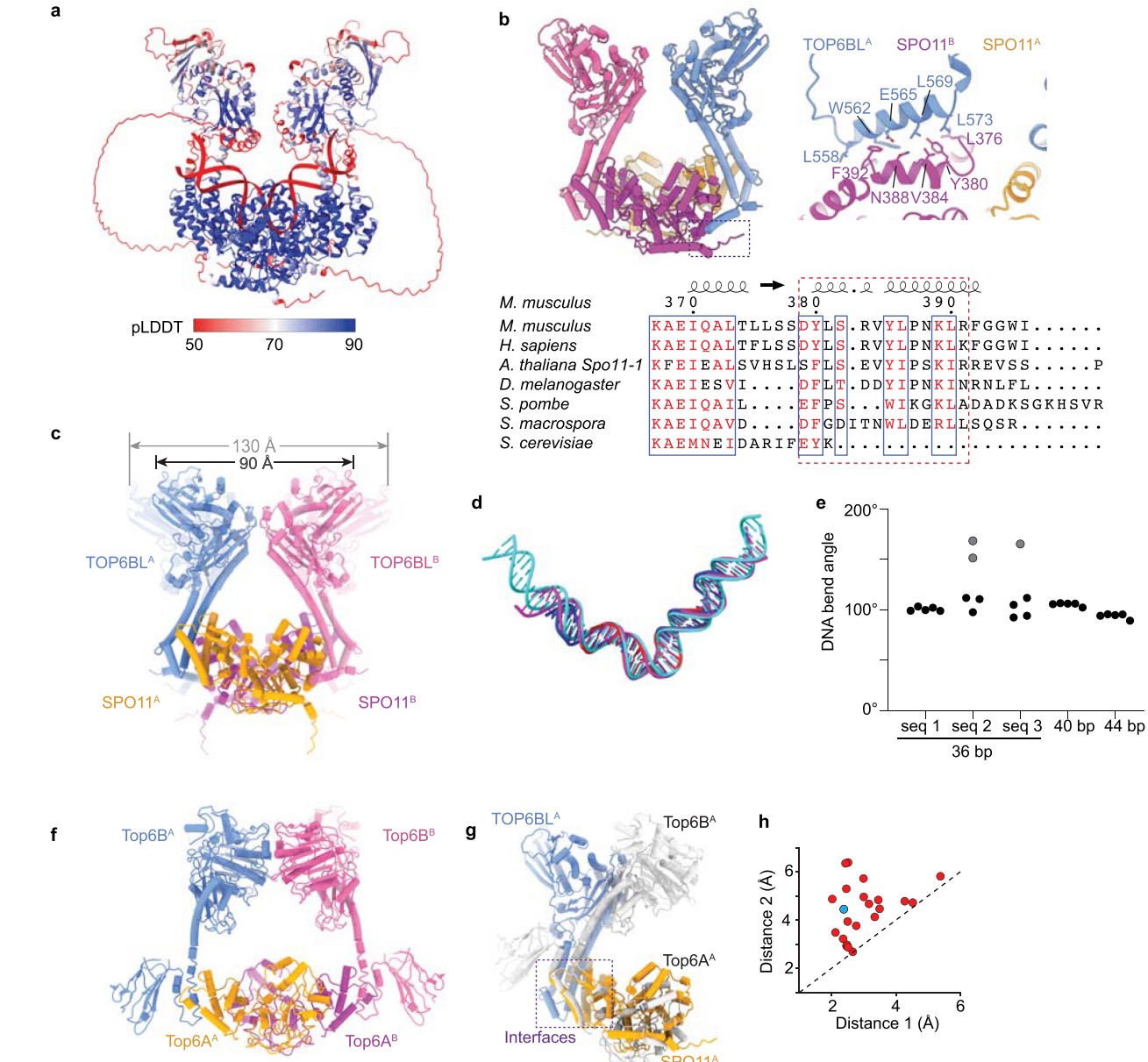

**Extended Data Fig. 3 | Computational models of SPO11–TOP6BL complexes bound to DNA. a**, AlphaFold3 model colored according to pLDDT confidence score, including unstructured segment and C-terminal helix from TOP6BL. **b**, Details of SPO11 interaction with the TOP6BL C terminal helix, predicted by many of the AlphaFold3 models (Supplementary Discussion 2). The boxed region in the overview (upper left) is detailed at upper right. The sequence alignment below shows conservation of this portion of SPO11 (red dashed box), but not in *S. cerevisiae*. **c**, Range of distances between TOP6BL GHKL domains for the models in Supplementary Fig. 2. **d,e**, Reproducibility of DNA bending in models with DNAs of different sequence and/or length. Duplexes are superimposed for three models (**d**), and bend angles from all 25 models are summarized (**e**). Three models with minimally bent DNA are shown in gray. Although the bend position varies considerably (causing the poor DNA pLDDT

score in panel **a**), the arm positions, bend angles, and local DNA deformation at the bend were all highly reproducible. **f**, Crystal structure of Topo VI holoenzyme from *Methanosarcina mazei* (pdb: 2q2e)[14]. **g**, Comparison of SPO11–TOP6BL and Top6A–Top6B (pdb: 2q2e) interfaces. **h**, Asymmetry of AlphaFold3 models. Each point indicates the distances in one AlphaFold3 model between the side chain oxygens of the two SPO11-Y138 residues and their respective "scissile" phosphates as defined by the center of the DNA bend and the offset needed to generate a two-nucleotide 5′ overhang. Distance 1 is the shorter of the two distances for each model. The cyan point is from the representative model shown in detail views throughout this paper. The dashed diagonal indicates expectation for perfect rotational symmetry. Values are omitted (off scale) for the three models with unbent DNA (gray circles in panel **d**).

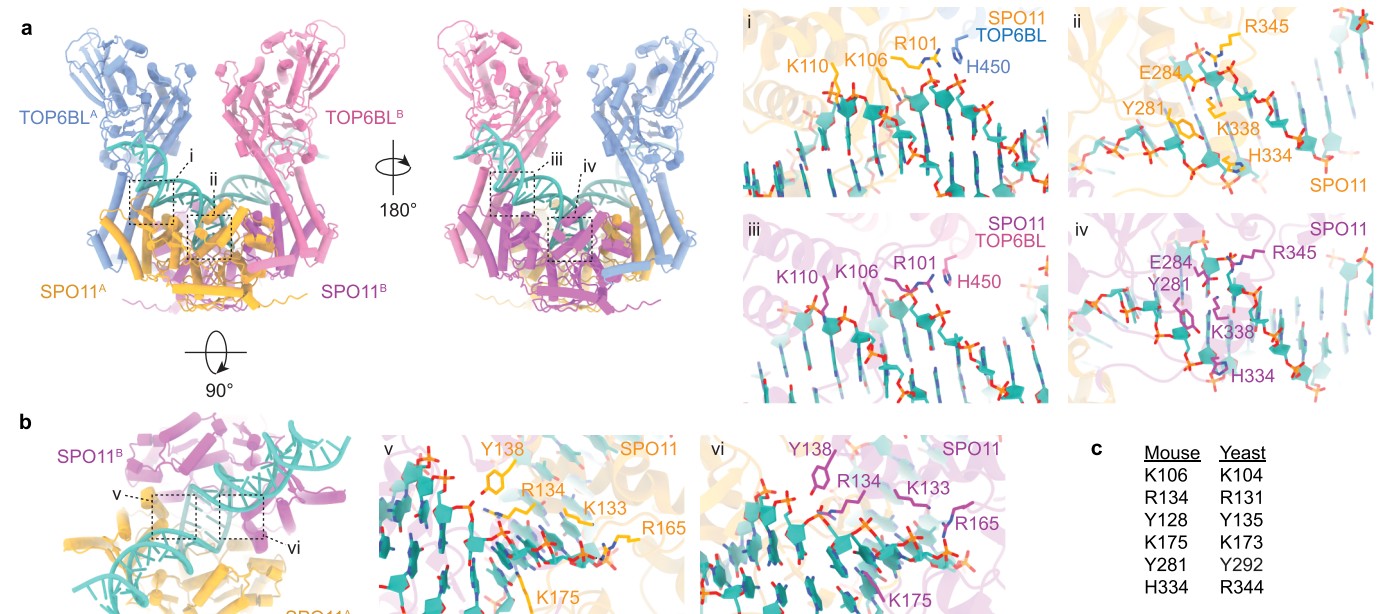

**a**

**b**

**c**

| Mouse | Yeast |
| --- | --- |
| K106 | K104 |
| R134 | R131 |
| Y128 | Y135 |
| K175 | K173 |
| Y281 | Y292 |
| H334 | R344 |

**Extended Data Fig. 4 | Protein-DNA interfaces. a**,**b**, Details of predicted protein-DNA interfaces. Panel **a** shows Patch A (panels ii and iv) and Patch B (panels i and iii) from Fig. 3d. Panel **b** focuses on contacts close to the DNA bend, scissile phosphates and active sites. **c**, Correspondence between DNA-contacting residues predicted by the mouse SPO11 AlphaFold3 model and observed in *S. cerevisiae* Spo11 cryo-EM structures[24].

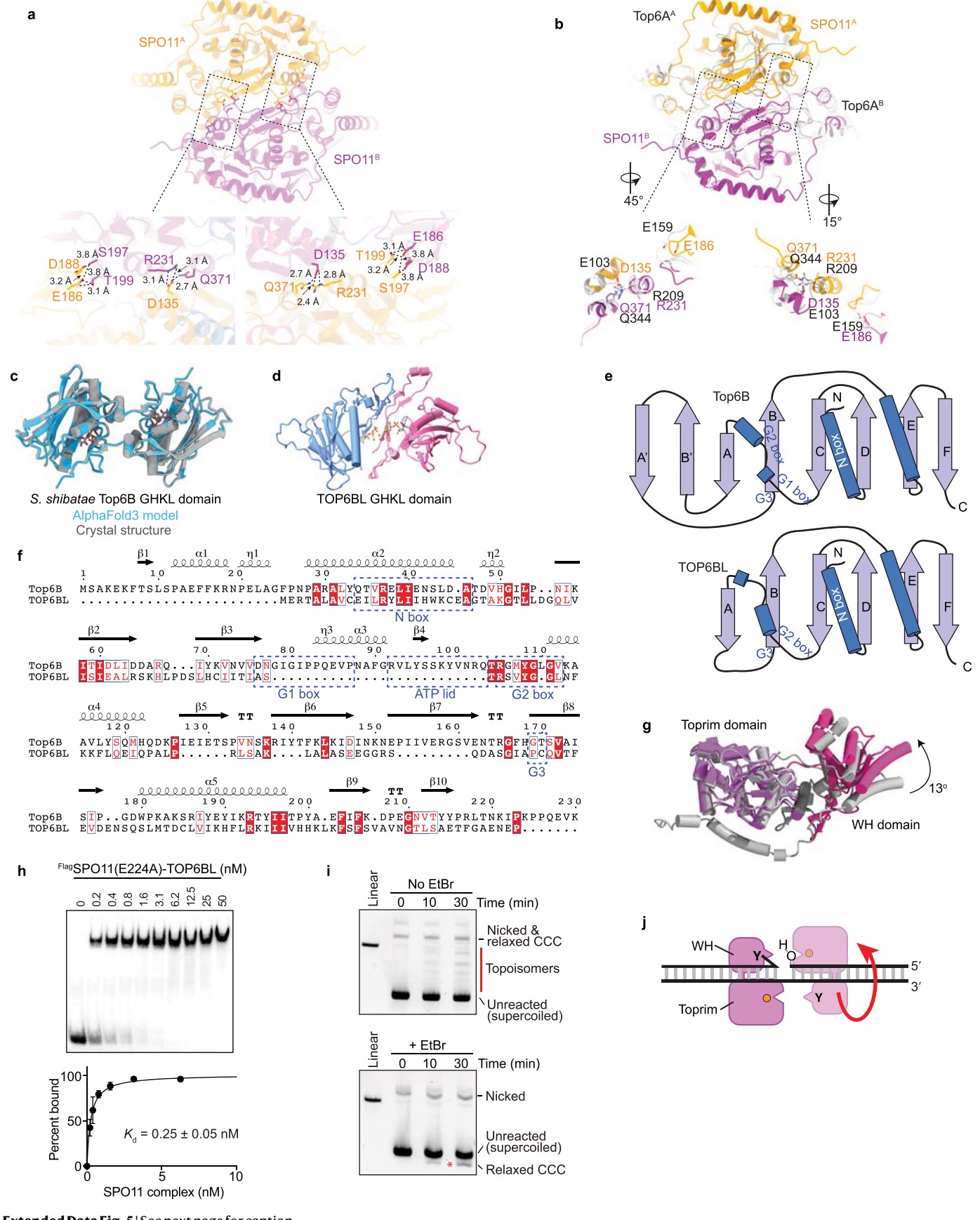

**Extended Data Fig. 5** | See next page for caption.

**Extended Data Fig. 5 | Analysis of the SPO11 dimer interface and TOP6BL GHKL domain. a,b**, SPO11 dimer interface. Panel **a** shows distances between conserved interfacial residues; panel **b** compares the SPO11 and Top6A dimer interfaces (*M. mazei* Topo VI; pdb: 2q2e)[14]. **c**, Agreement between AlphaFold3 model and crystal structure (pdb: 1z5a)[32] for an ATP-mediated dimer of the *S. shibatae* Top6B GHKL domain. **d**, Failure of AlphaFold3 to predict correct ATP-dependent dimerization of the TOP6BL GHKL domain. The dimer interface is in the wrong location and the ATPs are not at the correct surface. **e**, Comparison of Top6B and TOP6BL GHKL domain topologies. β strands (arrows) and α helices (cylinders) are shown, along with conserved ATP-interacting elements highlighted in panel **f**. **f**, Structure-based sequence alignment between *S. shibatae* Top6B and mouse TOP6BL GHKL domains. A previous alignment suggested that TOP6BL has degenerate versions of all three G boxes[9], but AlphaFold3 suggests that the G1 box is missing instead, with degenerate versions of the other two boxes plus the previously shown absence of the ATP lid[61]. **g**, Hypothetical conformation change between pre-DSB and post-DSB SPO11 complexes. A post-cleavage state was modeled by aligning the WH (hot pink) and Toprim (purple) domains separately to the orthologous domains in the yeast Spo11 cryo-EM structure[24]. The AlphaFold3 model (presumptive pre-cleavage state) is shown in gray. **h**, EMSA of E224A mutant SPO11 complexes binding to a 5′-labeled 25-bp hairpin substrate with a two-nucleotide 5′ overhang end. A representative gel is shown at left, quantification is at right (mean ± s.d. of n = 3 experiments; apparent $K_d$ given as mean ± s.e.). Retention of DNA binding is unlike the equivalent yeast Spo11-E233A[25]. **i**, Supercoil relaxation by wild-type protein. Wild-type SPO11–TOP6BL complexes were incubated with supercoiled pUC19 and 20 mM $CaCl_2$ then deproteinized and separated on agarose gels without (top) or with (bottom) 1 μg/ml ethidium bromide (EtBr). Red line, topoisomer ladder that disappears on EtBr-containing gels. Red asterisk, plasmids that were relaxed covalently closed circles (CCC) after the reaction and became positively supercoiled upon binding EtBr. Performed twice with similar results. **j**, Model for the mechanism of supercoil relaxation. The cartoon (not to scale) shows the arrangement of the WH and Toprim domains of each SPO11 monomer on nicked DNA, with the 5′ end covalently attached to Y138 of the darker pink monomer on the left, and the 3′ OH bound by the Toprim domain of the lighter colored monomer on the right. The red arrow signifies rotation of the right-hand SPO11 and the DNA arm it is bound to, relative to the left-hand monomer and its bound DNA arm. This rotation, which necessitates disruption of the SPO11 dimer interface, would allow the DNA to swivel around the uncut strand, thereby relaxing supercoils. See Supplementary Discussion 3 for more detail.

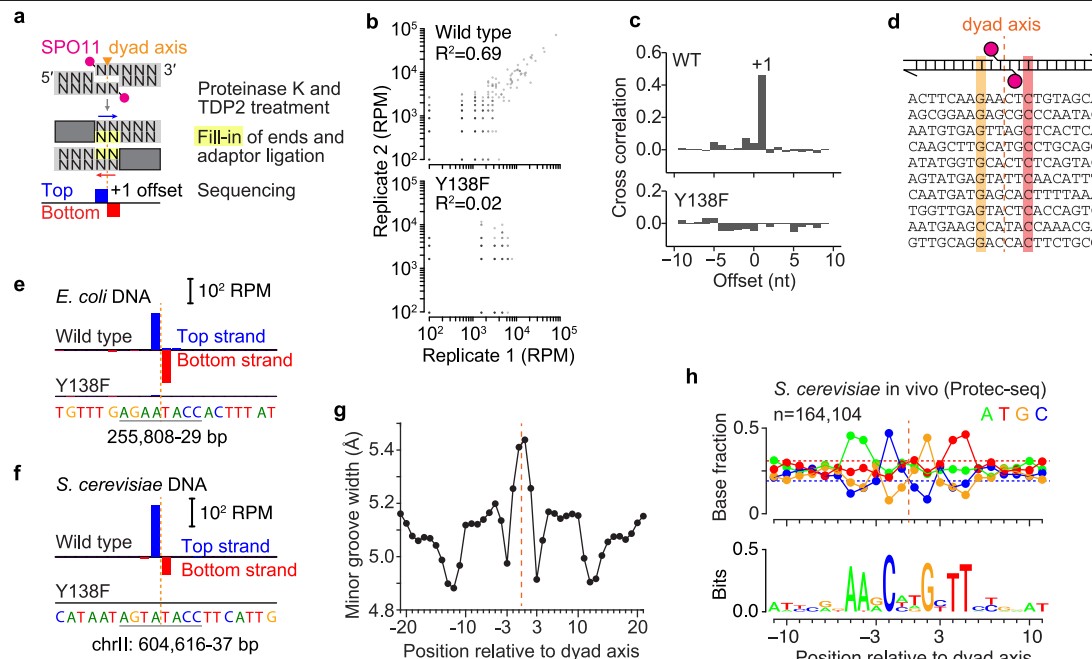

**Extended Data Fig. 6 | Analysis of SPO11 sequence preferences. a**, Schematic description of tyrosyl phosphodiesterase 2 (TDP2)-seq. Covalently bound protein was removed with proteinase K and TDP2, DNA ends were blunted with T4 DNA polymerase, adaptors were ligated to the blunted ends, and the ligation junctions were sequenced[36]. **b**, Correlation between replicate TDP2-seq datasets for in vitro cleavage of pUC19 with either wild-type (top) or Y138F mutant (bottom) SPO11 complexes. Each point is the read count for a nucleotide position in the plasmid. Positions with no TDP2-seq tags (n = 5,084 and 5,211 for wild type and Y138 mutant maps, respectively) were not used for calculating $R^2$, but were set as $10^{-2}$ for plotting purposes. **c**, Cross correlation between all top- and bottom-strand reads for in vitro cleavage of pUC19 with wild-type (WT) or Y138F mutant SPO11 complexes. **d**, Sequence context around the top 10 preferred cleavage sites on pUC19 mapped by TDP2-seq. The bases at −3 and +3 relative to

the dyad axis are highlighted. **e**,**f**, Examples of preferred in vitro cleavage positions on *E. coli* (**e**) or *S. cerevisiae* (**f**) genomic DNA, presented as in Fig. 4b except that read count is given in RPM. Orange dashed lines indicate the inferred dyad axis of cleavage. **g**, Example of DNA shape properties predicted for the base composition preferred by SPO11. Minor groove width was predicted[62,63] from the mononucleotide frequencies for preferred cleavage sites (n = 5180) on yeast genomic DNA. **h**, Base composition bias for *S. cerevisiae* Spo11 in vivo. DNA DSB ends generated by Spo11 (n = 164,104 sequenced DNA fragments) were identified in Protec-seq maps from *sae2/com1* mutants, in which DSBs remain unresected (data from[38]). Similar base composition biases were also reported for maps of Spo11 oligonucleotides[27,37]. Fractional base composition (above) and sequence logo (below) are shown.

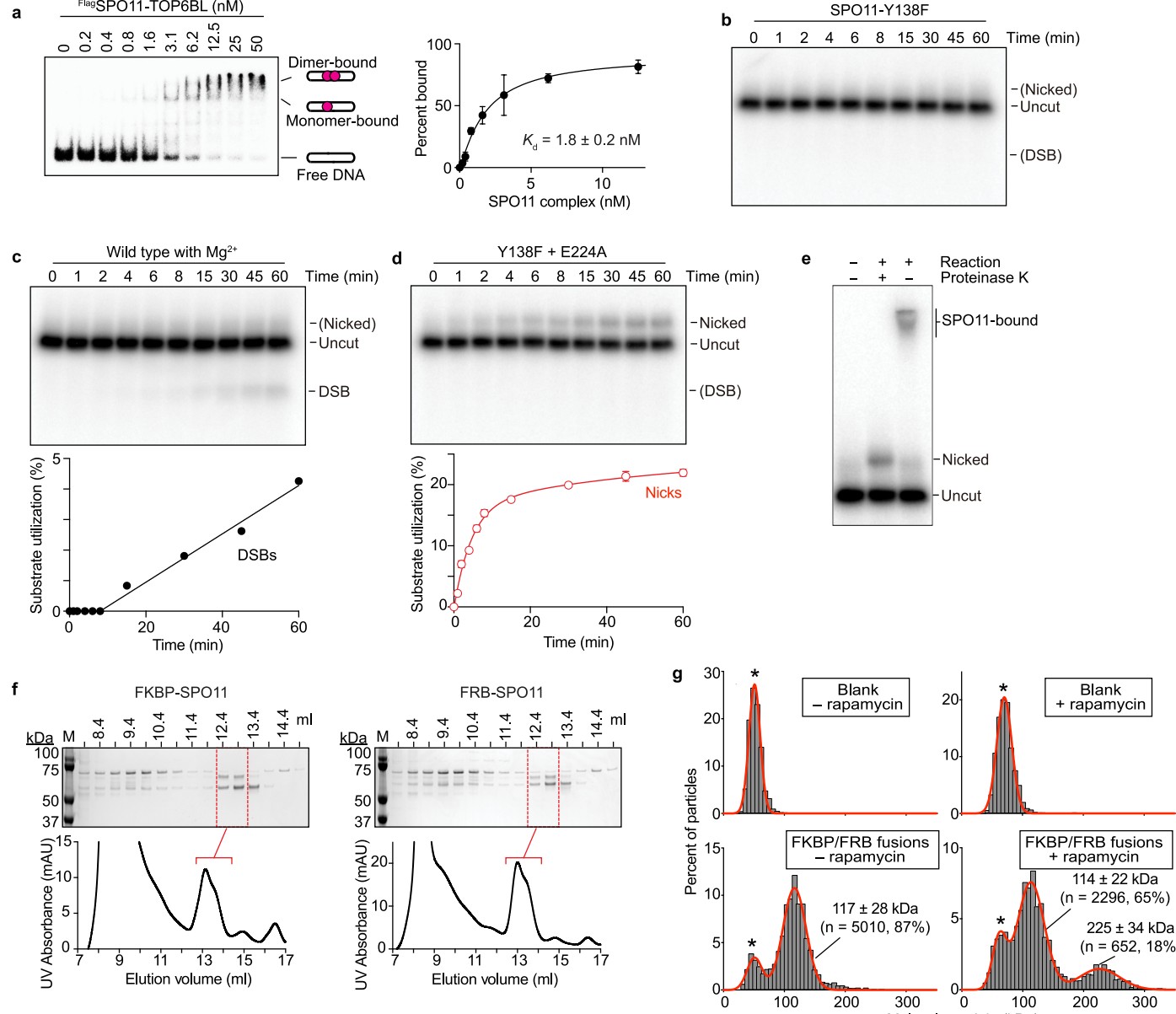

**Extended Data Fig. 7 | Enhancing SPO11 dimerization stimulates DNA cleavage activity. a**, EMSA of binding to the $^{32}$P-labeled oligonucleotide shown in Fig. 4f. A representative gel is shown at left, quantification at right (mean ± s.d. of n = 3 experiments). The apparent $K_d$ (mean ± s.e.) can be viewed as an estimate of the affinity of binding for the first monomer complex. The cartoons are for illustration; we do not know the exact conformation(s) of doubly-bound complexes. **b**, No cleavage of the oligonucleotide (0.5 nM) by SPO11-Y138F complexes (10 nM). Performed once. **c**, Weak cleavage supported by Mg$^{2+}$. Reactions contained 0.5 nM oligonucleotide, 10 nM wild-type SPO11 complexes, and 5 mM MgCl$_2$. Denaturing PAGE gel and quantification are provided (performed once). **d**, Nicking-only activity from mixture of Y138F and E224A mutant SPO11 complexes. Reactions contained 0.5 nM oligonucleotide, 5 nM of each mutant protein complex, and 5 mM MnCl$_2$. A representative gel is shown above, quantification is below (mean ± s.d. of n = 3 experiments). **e**, SPO11

covalently bound to nicked DNA. Reactions as in panel **d** were electrophoresed with or without prior digestion with proteinase K. Performed once. **f**, SEC profiles of FKBP-SPO11 or FRB-SPO11 complexes with TOP6BL. Coomassie-stained SDS-PAGE gels (above) and UV profiles (below) are shown for chromatography of anti-Flag affinity-purified material. Pooled fractions are indicated in red. This purification was conducted once. **g**, Mass photometry of purified FKBP-SPO11 and FRB-SPO11 complexes with TOP6BL. The blanks lacked protein; protein concentration in the lower graphs was 15 nM total (7.5 nM each fusion protein). Rapamycin was 5 μM when included. Particle counts (gray bars), gaussian fits (red lines), fitted mean ± s.d., and percentages of total particles are shown. Calculated masses are 123.0 kDa (FKBP) and 122.5 kDa (FRB) for monomeric and 245.5 kDa for dimeric complexes. Asterisks, background material also present in the blanks.

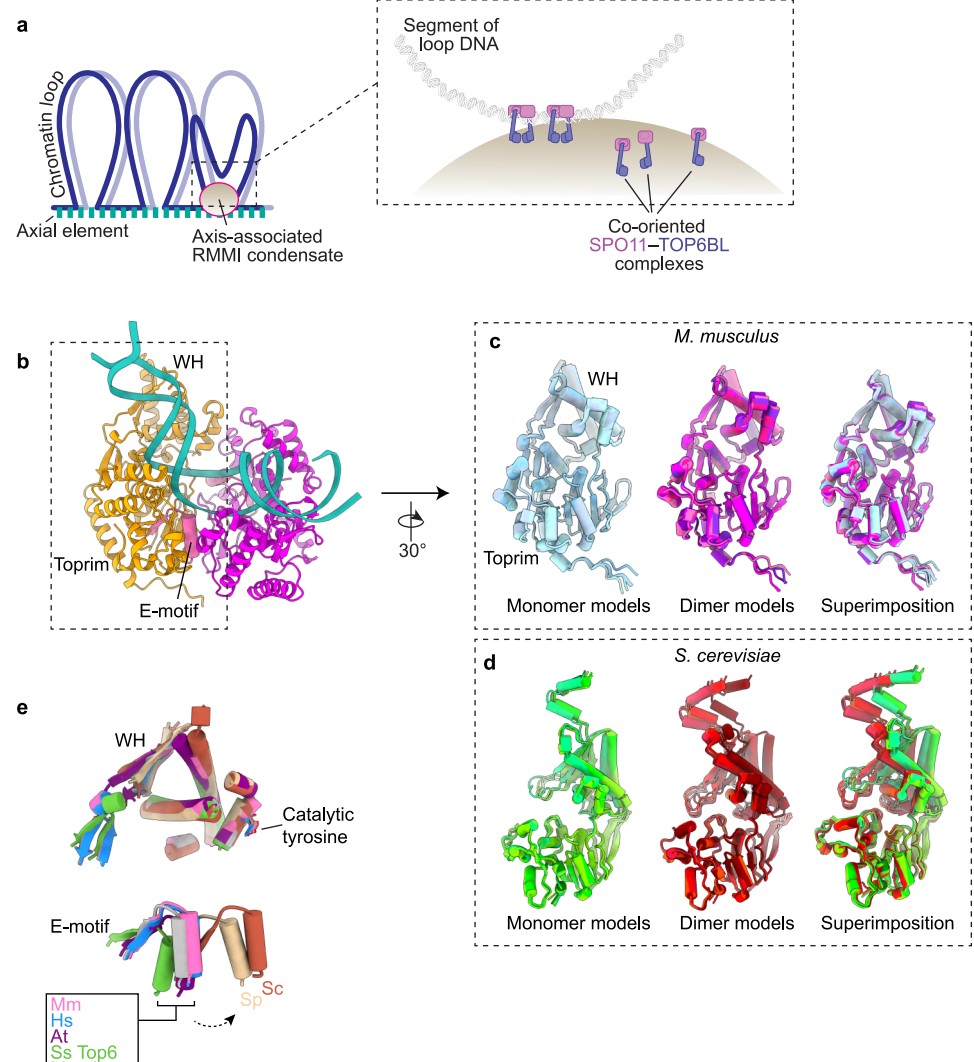

**Extended Data Fig. 8 | A low propensity for dimerization makes SPO11 dependent on accessory factors for activity in vivo. a**, Assembly of the DSB-forming machinery integrated with higher order chromosome structure. The cartoon on the left illustrates the organization of a meiotic chromosome as a linear axial element from which chromatin emanates in loops, with light and dark blue lines representing aligned sister chromatids[7]. In yeast, Rec114, Mei4, and Mer2 proteins assemble cooperatively with DNA and are thought to form axis-associated biomolecular condensates[42]. The mouse orthologs REC114, MEI4, and IHO1 (along with MEI1, which is not found in yeast) likely do so as well (RMMI)[64,65]. The cartoon at right illustrates the hypothesis that these condensates recruit and co-orient clusters of SPO11 core complexes, which can then form dimers to capture and ultimately break a segment of DNA from a nearby chromatin loop[66]. Figure adapted from ref. 24 under a CC BY license. **b–d**, SPO11 conformations in monomer and dimer models. Panel **b** presents an overview of the AlphaFold3 prediction of the *M. musculus* SPO11 dimer structure with each SPO11 chain colored as Fig. 3a. The E-motif (residues 219–235) from the orange chain is indicated; this segment from the Toprim domain contains the conserved E224 predicted to bind magnesium. Panels **c** and **d** compare the conformations of individual SPO11 chains from AlphaFold3 models of monomers (left, top 5 ranked models) or dimers (middle, top 5 ranked models) for mouse (**c**) or *S. cerevisiae* (**d**). As shown in the superimposed images at right, the monomer and dimer models are similar for the mouse protein but are distinct for yeast, with different relative conformations of the WH domain relative to the Toprim domain. (The dark red dimer model (**d**, middle) predicts an incorrect DNA position, so it is excluded from the monomer-dimer comparison (**d**, right)). **e**, Comparison of predicted positions of the Toprim domain relative to the WH domain for SPO11 homologs from various species. To evaluate the evolutionary conservation of the distinct monomer conformation in panel **c**, SPO11 monomer structures predicted by AlphaFold3 were compared for *M. musculus* (Mm, hot pink), *Homo sapiens* (Hs, light blue), *Arabidopsis thaliana* (At, Spo11-1, purple), *S. cerevisiae* (Sc, brown), *Schizosaccharomyces pombe* (Sp, tan), and *Sulfolobus shibatae* (Ss, Top6A, green). We simplified the display of the WH vs. Toprim domains by aligning on the WH domain (only partial secondary structures of WH used for alignment are shown for simplicity), then showing just the E-motif positions as an indication of the relative position of the Toprim domain. For reference, segments from the mouse SPO11 dimer model are shown in gray. Notably, the monomer structures from *S. cerevisiae* and *S. pombe* exhibit distinct E-motif positions from the other homologs. It is not clear why AlphaFold3 makes different predictions for the yeast protein monomers, but we speculate that it reflects an inherent tendency of the yeast proteins to adopt a conformation that may be less compatible with dimerization.

# Reporting Summary

## Statistics

For all statistical analyses, confirm that the following items are present in the figure legend, table legend, main text, or Methods section.

| n/a | Confirmed | |
|---|---|---|
| ☐ | ☒ | The exact sample size (*n*) for each experimental group/condition, given as a discrete number and unit of measurement |
| ☐ | ☒ | A statement on whether measurements were taken from distinct samples or whether the same sample was measured repeatedly |
| ☒ | ☐ | The statistical test(s) used AND whether they are one- or two-sided *Only common tests should be described solely by name; describe more complex techniques in the Methods section.* |
| ☒ | ☐ | A description of all covariates tested |
| ☒ | ☐ | A description of any assumptions or corrections, such as tests of normality and adjustment for multiple comparisons |
| ☐ | ☒ | A full description of the statistical parameters including central tendency (e.g. means) or other basic estimates (e.g. regression coefficient) AND variation (e.g. standard deviation) or associated estimates of uncertainty (e.g. confidence intervals) |
| ☒ | ☐ | For null hypothesis testing, the test statistic (e.g. *F*, *t*, *r*) with confidence intervals, effect sizes, degrees of freedom and *P* value noted *Give P values as exact values whenever suitable.* |
| ☒ | ☐ | For Bayesian analysis, information on the choice of priors and Markov chain Monte Carlo settings |
| ☒ | ☐ | For hierarchical and complex designs, identification of the appropriate level for tests and full reporting of outcomes |
| ☒ | ☐ | Estimates of effect sizes (e.g. Cohen's *d*, Pearson's *r*), indicating how they were calculated |

*Our web collection on statistics for biologists contains articles on many of the points above.*

## Software and code

Policy information about availability of computer code

| Data collection | Mass Photometry: Refeyn AcquireMP (version 2024.1.1.0) |
|---|---|
| | Atomic force microscopy: JPK Scanning Probe Microscope Control Program (version 8.0.59.1) |
| | Phosphor Imaging scan: Amersham typhoon control software (version 4.0.0.4) |
| | SYBR Gold Scan: BioRad Image Lab Touch Software (version 3.0.1.14) |
| | Illumina sequencing: Real Time Analysis (version 3.1 or version 4.1) |
| | Size exclusion chromatography: UNICORN 7.6 (version Build 7.6.0.1306) |
| | A260/280 for protein quantification: NanoDrop 8000 (version 2.3.3) |
| | AlphaFold3 (server: https://alphafoldserver.com) |
| | UNICORN 7.6 (version Build 7.6.0.1306) |
| | NanoDrop 8000 (version 2.3.3) |
| | Amersham typhoon control software (version 4.0.0.4) |

| Data analysis | GelBandFitter (version 1.7) |
|---|---|
| | GraphPad Prism 10 (version 10.3.0 (461) for Mac OS X) |
| | Image Lab (version 6.1.0 build 7) |
| | ImageJ (version 1.54g) |
| | DRAGEN suite (version 4.2.7) |
| | bowtie2 (version 2.5.3) |
| | ggseqlogo (version 0.2) |
| | European Molecular Biology Open Software Suite [EMBOSS] (version 6.6.0) |
| | R (versions 4.2.3 and 4.3.2) |

Chimera (version 1.18)
ChimeraX (version 1.8)
Refeyn DiscoverMP (version 2024.1.0.0)
JPK Data Processing Software (version 8.0.59.1)

For manuscripts utilizing custom algorithms or software that are central to the research but not yet described in published literature, software must be made available to editors and reviewers. We strongly encourage code deposition in a community repository (e.g. GitHub). See the Nature Portfolio guidelines for submitting code & software for further information.

# Data

Policy information about availability of data

All manuscripts must include a data availability statement. This statement should provide the following information, where applicable:
- Accession codes, unique identifiers, or web links for publicly available datasets
- A description of any restrictions on data availability
- For clinical datasets or third party data, please ensure that the statement adheres to our policy

Raw and processed TDP2-seq data are available at GEO under accession number GSE275291 (https://www.ncbi.nlm.nih.gov/geo/query/acc.cgi?acc=GSE275291). The AlphaFold3 model used to generate most of the figures is provided in .pdb format as Supplemental Data. The mouse genome assembly mm10 (a.k.a. GRCm38) is available at https://www.ncbi.nlm.nih.gov/datasets/genome/GCF_000001635.20/

# Research involving human participants, their data, or biological material

Policy information about studies with human participants or human data. See also policy information about sex, gender (identity/presentation), and sexual orientation and race, ethnicity and racism.

| | |
|---|---|
| Reporting on sex and gender | n/a, no human subjects data |
| Reporting on race, ethnicity, or other socially relevant groupings | n/a, no human subjects data |
| Population characteristics | n/a, no human subjects data |
| Recruitment | n/a, no human subjects data |
| Ethics oversight | n/a, no human subjects data |

Note that full information on the approval of the study protocol must also be provided in the manuscript.

# Field-specific reporting

Please select the one below that is the best fit for your research. If you are not sure, read the appropriate sections before making your selection.

☒ Life sciences  ☐ Behavioural & social sciences  ☐ Ecological, evolutionary & environmental sciences

For a reference copy of the document with all sections, see nature.com/documents/nr-reporting-summary-flat.pdf

# Life sciences study design

All studies must disclose on these points even when the disclosure is negative.

| | |
|---|---|
| Sample size | All DNA cleavage experiments were performed in triplicate unless stated otherwise. This sample size was chosen because preliminary experiments determined it to be sufficient to distinguish effect sizes deemed to be meaningful for purposes of the studies presented. Sequencing experiments were conducted in two replicates (biological replicates for in vivo data). Prior published work (https://pubmed.ncbi.nlm.nih.gov/39149289/) has established that two replicates are sufficient to determine reproducibility because of the high correlation between replicates for these methods. |
| Data exclusions | For sequencing data: sequence reads mapping at chromosome ends were excluded because they reflect natural end structure and not SPO11-dependent cleavage; sequence reads mapping to repetitive DNA were excluded because they cannot be assigned with confidence to specific locations, and our previous published work has established that excluding them does not affect the conclusions drawn. No other data were excluded. |
| Replication | All in vitro DNA cleavage experiments were repeated at least twice with similar results. The protein purifications were repeated at least twice, and for wild type proteins more than five times each, with similar results. DNA sequencing experiments were performed in two replicates each and assessed for reproducibility by direct quantitative comparison; all sequencing experiments were successfully replicated. |
| Randomization | Experiments involved comparison of mutants to matched wild-type controls, or comparison of the same protein's behaviors under different reaction conditions, so randomization is neither necessary nor appropriate. |

| Blinding | No blinding was used. All biochemical experiments involved comparison of wild type with mutant proteins, or comparison of different in vitro conditions for the same protein. In vivo experiments involved independent replicates performed on mice of the same genotype. It is not standard practice in the field to use blinding for these assays. Moreover, blinding is not necessary with this experimental design because meaningful effect sizes are larger than any likely effects of operator bias. |

# Reporting for specific materials, systems and methods

We require information from authors about some types of materials, experimental systems and methods used in many studies. Here, indicate whether each material, system or method listed is relevant to your study. If you are not sure if a list item applies to your research, read the appropriate section before selecting a response.

## Materials & experimental systems

| n/a | Involved in the study |
|-----|----------------------|
| ☐ | ☒ Antibodies |
| ☐ | ☒ Eukaryotic cell lines |
| ☒ | ☐ Palaeontology and archaeology |
| ☐ | ☒ Animals and other organisms |
| ☒ | ☐ Clinical data |
| ☒ | ☐ Dual use research of concern |
| ☒ | ☐ Plants |

## Methods

| n/a | Involved in the study |
|-----|----------------------|
| ☒ | ☐ ChIP-seq |
| ☒ | ☐ Flow cytometry |
| ☒ | ☐ MRI-based neuroimaging |

## Antibodies

| Antibodies used | anti-Flag M2 affinity gel (Sigma A2220)<br>anti-Flag-HRP monoclonal antibody (mouse, 1:1000, Sigma A8592)<br>anti-Flag magnetic agarose (Pierce A36797) |
| Validation | Antibody specificity was confirmed by data shown in the paper, namely, successful affinity purification of recombinant Flag-tagged SPO11 complexes in purifications and immunoprecipitations, and absence of detectable signal in negative controls in immunoblotting experiments. |

## Eukaryotic cell lines

Policy information about cell lines and Sex and Gender in Research

| Cell line source(s) | FreeStyle™ 293-F cells, Invitrogen |
| Authentication | The cell lines were not authenticated |
| Mycoplasma contamination | Cell lines were not tested for mycoplasma. |
| Commonly misidentified lines (See ICLAC register) | None |

## Animals and other research organisms

Policy information about studies involving animals; ARRIVE guidelines recommended for reporting animal research, and Sex and Gender in Research

| Laboratory animals | Mus musculus, young adult (<4 mos old) Mre11 conditional knockout mice (Mre11-flox/del Ngn3-Cre on a congenic C57BL/6J background) were used. |
| Wild animals | The study did not involve wild animals |
| Reporting on sex | Experiments analyzed spermatogenesis, so only male mice were used. |
| Field-collected samples | The study did not involve samples collected from the field. |
| Ethics oversight | Mouse experiments were performed in accordance with US Office of Laboratory Animal Welfare regulations and were approved by the Memorial Sloan Kettering Cancer Center Institutional Animal Care and Use Committee. |

Note that full information on the approval of the study protocol must also be provided in the manuscript.

# Plants

Seed stocks

*Report on the source of all seed stocks or other plant material used. If applicable, state the seed stock centre and catalogue number. If plant specimens were collected from the field, describe the collection location, date and sampling procedures.*

Novel plant genotypes

*Describe the methods by which all novel plant genotypes were produced. This includes those generated by transgenic approaches, gene editing, chemical/radiation-based mutagenesis and hybridization. For transgenic lines, describe the transformation method, the number of independent lines analyzed and the generation upon which experiments were performed. For gene-edited lines, describe the editor used, the endogenous sequence targeted for editing, the targeting guide RNA sequence (if applicable) and how the editor was applied.*

Authentication

*Describe any authentication procedures for each seed stock used or novel genotype generated. Describe any experiments used to assess the effect of a mutation and, where applicable, how potential secondary effects (e.g. second site T-DNA insertions, mosiacism, off-target gene editing) were examined.*