## [Peer Review file · Nature]

Reconstitution of SPO11-dependent double-strand break formation

Corresponding Author: Dr Scott Keeney

Parts of this Peer Review File have been redacted as indicated to maintain the confidentiality of unpublished data. This file contains all reviewer reports in order by version, followed by all author rebuttals in order by version.

Version 1:

Reviewer comments:

Referee #1

(Remarks to the Author)

The manuscript "Reconstitution of SPO11-dependent double-strand break formation" by Zheng...Keeney reports the reconstitution of SPO11-mediated DNA breaks in vitro, an achievement that can reasonably be called a holy grail of the meiosis field, and one that has been 27 years in the making. Because of its obvious importance and broad appeal, plus the fact that the experiments are all of exceptional quality and clearly explained, I strongly support publication after the authors address only a few questions and concerns, detailed below.

Fig. 1f: Can the authors comment on whether the observed SPO11 complexes in AFM are monomeric (1:1), dimeric (2:2), or larger aggregates, based on the size of the protein blobs bound to DNA?

I am as big an AlphaFold fan as you'll find. But, (like the authors I suspect), I do find the lack of two-fold symmetry in AF3 models of DNA-bound TOP6BL-SPO11 complex to be... weird. Usually AlphaFold seems to go out of its way to predict perfectly symmetric complexes. I'm not going to suggest taking this analysis out, but can the authors add a bit more explanation regarding the reproducibility of the exact mode of DNA binding depicted in Fig. 3d across the many models they ran?

On the topic of AlphaFold, were all predictions run with preferred DNA sequences according to the preferences described in Fig. 4? If so, what happens with DNA sequences that are designed specifically to be non-preferred?

The prevalence of nicks in the biochemical experiments is interesting, and I have a few questions on this topic. First, the prevalence of nicks generated by dimeric complexes in vitro suggests that nicking may occur at a relatively high frequency in vivo. Is there any way (in existing datasets) to support this idea? For future work, would there be any way to detect nicks generated by SPO11 complexes in cells?

In Fig. 3h, can the authors explain why Unreacted (supercoiled) and Relaxed CCC run differently on a gel with EtBr? Aren't they supposed to be the same molecule (chemically), and doesn't EtBr saturate the DNA enough to positively supercoil even DNA that (as purified from bacteria) is negatively supercoiled?

I would recommend caution in interpreting the finding that the SPO11-TOP6BL complex can mediate topoisomerase-like activity in vitro. As the authors rightly point out, this activity seems limited to the cases where the complex generates a nick, and therefore the dimer is kept close together even while one strand is broken. Given the chemistry of the reaction, it's almost inevitable that relegation can happen in that case (sometimes with relaxation via rotation about the remaining strand, a la Topo I). This is almost certainly not true "type II topoisomerase" strand passage, and as such I would urge restraint when describing. I think the authors did a good job of this, except I would suggest leaving this result out of the abstract.

Does LZ-SPO11 purify as a dimeric complex?

Does the LZ-SPO11 complex mediate topoisomerase-style activity? If the dimer interface is stable, it should not (if the authors' model is correct).

Do the authors anticipate that forcing dimerization of the yeast Spo11 core complex (as in the LZ-SPO11 fusion) would

enable it to catalyze DSB formation in vitro?

For AF3 discussions (and possibly other places) - please clarify if rmsd's are for all atoms or only C-alpha atoms (C-alpha rmsd is typically used for these kinds of comparisons).

Finally - congratulations to the authors on this achievement!

Referee #2

(Remarks to the Author)

It has long been acknowledged that the SPO11 protein, related to TopoVI, is the catalytic subunit responsible for DSB formation during meiosis, but for 27 years there had been no demonstration that SPO11 can make DSBs in vitro. With this paper and an accompanying paper from Oger and Claeys Bouuaert, the long wait is over. Using the mouse SPO11-TOP6BL dimer, the authors have detected several activities in addition to the DNA-binding activity already shown for the yeast protein: double-strand cleavage; single-strand nicking; and supercoil relaxation. Based on previous structural data and on structural modeling, the authors predict a pre-cleavage complex that contains two SPO11-TOP6BL dimers bound to bent DNA, with the two active sites formed by contributions by both SPO11 monomers. The activity is very inefficient, which is attributed to a combination of inefficient dimer formation and only a minority of dimers having an active conformation. Support for some of these conclusions come from an inter-subunit complementation experiment and from an artificial tethering experiment. In vitro and in vivo cleavage site sequence preferences are similar, supporting the idea that the activities seen in vitro recapitulate the in vivo activity, but this is the extent of the validation. In concluding, authors emphasize the idea that the monomer-dimer dichotomy is important for the regulation of SPO11 activity, with other proteins of the DSB complex, known to form condensates, providing the environment to concentrate Spo11 and promote formation of active dimers.

This is an important advance in the field, and it almost certainly will be cited in every paper on meiotic recombination for the foreseeable future. It opens the door for further biochemical characterization of SPO11's cleavage activity and its regulation. Because the conclusions rely exclusively on behavior of the complex in vitro, and on in silico modeling of the complex's structure, it will be important in the future to provide in vivo validation of the conclusion and to obtain real structures of the proteins and substrate. Now that the door has been opened, it is likely that many will be rushing through.

Comments:

1. It is somewhat disappointing that tethering the two SPO11 subunits resulted in only about a two-fold increase in initial rates, somewhat undercutting suggestions of the importance of the monomer-dimer transition. These experiments were performed at 75 mM SPO11, which is close to the optimal concentration for activity of the untethered complex. Is it possible that, by decreasing the SPO11 concentration and/or increasing the substrate concentration, a greater effect might be observed?
2. As mentioned above, SPO11 has single-strand nicking, single-strand resealing, and double-strand cutting activity. By comparing in the same experiment the kinetics of nick formation, DSB formation, and supercoil relaxation, could one come up with a possible model for how the reaction proceeds? At a minimum, it would be useful to provide separate plots for nicking and cutting, rather than just a single line for "substrate utilization" (Figs. 2a,h, 3g, etc.).
3. In Fig. 2d, only linear product is immunoprecipitated, but one would expect both linear and nicked products to be covalently linked to SPO11. Perhaps this is because the starting material contained a vast excess of linear over nicked products. It's important to start with material that has both nicked and linear products in it (perhaps an earlier time point, or a reaction run in Mg⁺⁺, or one run at lower temperature?).
4. Line 315 and following: The logic as to why stronger sequence signatures are seen with more complex substrates could have been presented more clearly. Is it possible that this is just a result of the lower number of unique reads with pUC19?
5. While the in silico structural modeling is voluminous, it is limited to comparisons between mouse and yeast. Could more of value be learned by extending the comparisons to other SPO11s?

Referee #3

(Remarks to the Author)

Zheng et al. reconstituted SPO11-TOP6BL complex and characterized its DNA cleavage activity in vitro. As revealed during meiosis, SPO11-TOP6BL complexes cleave DNA to form covalent 5' attachments and this process requires SPO11 active site residues, divalent metal ions, and SPO11 dimerization. Additionally, SPO11 also showed topoisomerase activity to relax DNA supercoils and reseal nicked DNA. AlphaFold modeling and deep sequencing of in vitro cleavage products further provided valuable information for SPO11 complex. These results are very interesting. This ms is well written. I'd like it.

It is necessary to have a comprehensive discussion of the cleavage differences (target DNA and SPO11 complex activity) observed from in vitro and in vivo experiments. It would be better to compare/discuss the conservation and difference for SPO11 complex in different organisms, such as yeast, human, and mouse (maybe with the help of AlphaFold prediction?).

1. Lines 103-104: Cite the figure (Fig. 1d, 25 and 50 nm), and it would be helpful to show the size markers on the gel in Fig. 1d, and also other gels.
2. Lines 116-117: Does this mean the structural difference between yeast and mouse? If so, what difference?
3. Lines 161-162: Negative supercoiled DNA was cut more efficiently; what is the interpretation for this? Actually in yeast, it seems more DSBs and COs form in negative supercoil rich regions. Are they related?
4. Lines 133-234: Would you please clarify the conformational change?
5. Lines 343-344: Is the preferred base composition related with the formation of negative supercoils or the organization of chromosome loop/axis?
6. Fig. 1d,e: Size markers for the gels would be helpful; the number of repeats of experiments and error bars are needed in 1e legends. 1f, what about distance between the two binding sites? 1g, cannot see the DNA alone data, please use another color?
7. Fig. 2a: Linear indicates a single cleavage site on the plasmid? what are those bands (obviously in WT at higher concentrations) between linear and supercoiled bands? Please clarify. 2d, a little bit confused: 2a showed that most supercoiled plasmids are cut once and thus predominantly linear ones. However, here, 2d showed that supercoiled plasmids are cut more than once and thus multiple discrete bands. if so, what caused the difference?
8. Fig. 4g: Please also show the DNA markers/ sizes for the gel; bottom quantification, not sure Error bars are shown, at least they are invisible in nicks.
9. ED Fig. 2e: Negatively supercoiled.
10. ED Fig. 6a-d: Size markers needed.
11. The manuscript would be further improved if they can compare/discuss the conservation and difference for SPO11 complex in different organisms, such as yeast, human, and mouse (maybe with the help of AlphaFold prediction?).

Referee #4

(Remarks to the Author)

This manuscript reports the characterization of the DNA cleavage activity by the mouse Spo11/Top6bl complex.

This is a very important and remarkable achievement in the field of meiosis. Spo11 and then Top6bl were identified several years ago, with substantial evidence showing that Spo11 carries the catalytic activity. However, despite several purification attempts, using proteins from various species, no DNA cleavage was obtained.

Here the authors purify a complex of mouse Spo11 and Top6bl. They show that it forms a heterodimer. They analyse its DNA binding properties. They detect DNA cleavage (nicking and linearization) on plasmid DNA (various topological states, including linear), and crucially this activity is abolished in the Spo11Y138F mutant. They show that Spo11 is covalently attached to the 5' ends. Interestingly, by combining Spo11Y138F and Spo11E224 proteins, they detect nicking activity, which provides support for the active complex being a tetramer. Surprisingly, a DNA relaxation activity is detected. Based on AF3, some unanticipated features of the complex are detected (asymmetry at the dyad axis and DNA unwinding). By nucleotide resolution mapping of the cleavage sites, preference for some nucleotides is detected.

Altogether, this is a great piece of work; the data is very clear and the interpretations are convincing. As noted by the authors, there is one major missing property as compared to the predicted *in vivo* activity: the formation of the heterotetramer. This leaves an unknown, which is that double-strand DNA cleavage is detected without detection of the tetrameric form, and it indicates that a small proportion of the purified protein is active for DNA cleavage. This, however, does not lower the interest and significance of the data presented here. I definitively support the publication: it is a breakthrough in the field.

A few points require revisions at various levels.

The overall presentation is clear. Importantly, the methods are carefully presented, which is key specifically for the purification steps. Several clarifications, however, should be provided with respect to DNA binding, ATP binding and DNA cleavage.

1) Protein purification: The authors should discuss and address the question of whether the tag added (FLAG) may alter or not SPO11 conformation. Based on studies in yeast, it is known that Spo11 is sensitive to tag addition (especially at the C-ter). Since getting *in vivo* validation is not reasonable in the context of this study, one needs to know if a linker was added? Which one? What are the predicted consequences on the structure by modelling?

2) DNA binding: Why increased concentration (3.1 nM) restores double-end binding to all oligos is not clear.

With the data provided (EMSA and AFM), at the end it is not clear what is the DNA-end specificity of complex. Does the

complex preferentially bind to 5' 2-nt overhangs (as the yeast complex) compared to hairpins? It would be interesting to have a long DNA fragment (linearized plasmid) with hairpins at both ends (or relaxed DNA) to investigate duplex binding (by gel shift or AFM) (Yeh et al. (2017) detected discrete DNA-protein complexes with *C. elegans* Spo-11, by gel shift). In Fig. 1g, the quantification of bending should be clarified: This reviewer assumes that one-end, three- and four-way are excluded for the quantification and that molecules with internal binding on duplex are selected. Is this correct? Ideally, one would like to know if tetramers, even if low in proportion, could be identified. Potentially by immunogold labelling?

3) ATP binding: Whether Top6bl binds ATP or not is very important to know. The authors discuss this only in the context of the AF3 model. It would be interesting to know if they have assayed ATP binding activity of the complex even if it is probably not so exciting to present a negative result. In addition, they should at least refer to the two studies addressing this question on the Top6bl subunit alone: Diagouraga et al. (2024) and Chen et al. (2024, NAR). In addition, in Top6bl, it is not only the G1 box which is missing but also the G2 (which includes three conserved glycines) is altered (the first conserved glycine is substituted by a serine). The presentation of results (L212-221) should also refer to previous modelling showing that the ATP lid is missing (Nore et al., 2022), as well as in *A. thaliana* MTopVIBL (Vrylinck et al., 2016).

4) DNA cleavage: In the methods, the strategy used for quantification is described, it is, however, not fully clear; it is mentioned that "when multiple DSBs are clearly present, substrate utilization was quantified as disappearance of the supercoiled band". In which panels was this applied? Otherwise, it is the "fraction of DNA molecules that had sustained at least one detectable nick or DSB" (why "at least"? How are multiple DSB cleavages quantified here? and it is also written that cleavage is underestimated). It would be interesting to have the quantification of multiple cleavages (by quantifying the smear below the size of the linear product) when these products are prominent (such as in Fig. 2d).

One question that also arises is whether the kinetics of nicked and linear products are identical. Thus, each product should be quantified separately.

In the text, the results of panel 2d cannot be presented without explaining or describing the smear (multiple cuts) in the WT IP lane.

Overall, this reviewer understands that both nicks and DSBs most likely result from the complex in a tetrameric form but that nicks are abundant due to some instability or improper assembly of the tetramer (improper dimerization of Spo11). If this is correct, this should be more explicitly stated in the description of the results.

5) Complex assembly: Since most protein is assembled as monomer of Spo11/Top6bl, it is important to understand the dynamic of the activity. The experiment in Fig. 2h is not fully convincing: how do the authors exclude that the cleavage on the plasmid added second may be due to some free Spo11/Top6bl complex rather than resulting from slow exchange? The molar ratio complex/DNA is very high; do the authors imply that all or most of these complexes are bound from the start of the reaction? Whether the slope of the cleavage of the second plasmid is the same as the second phase of the first plasmid is not shown by the graphs presented. In the reaction there is probably a mix of monomers (in excess), improperly assembled dimers (may be leading to nicking), and properly assembled dimers (DSB cleavage), and this, in a dynamic status. Please clarify.

6) The nicking activity of the complex: The combination of Y138F and E224A is a great experiment showing the role of the expected Spo11 dimer. It is the only experiment allowing to conclude as written on L400, that Spo11 can dimerize on DNA. Evidence for topoisomerase activity with the WT protein should also be validated by EtBr addition (as shown for the mutants in 3h). This unexpected property could be discussed, and specifically, whether it is an in vitro artefact or whether it may have significance in vivo.

7) DNA sequence preference: The mapping of cleavage on pUC19 includes both single-strand nicks and double-strand cuts. It would be interesting to determine if there is any difference. It is surprising that the authors have used as genomic DNA *E. coli* and yeast DNA. It would be more logical to test mouse genomic DNA (and to use another DNA as carrier if needed for the library preparation). Is there any specific rationale for this choice?

8) The hairpin linear substrate: This is an important experiment but somewhat confusing, and several additional information are needed for interpreting this data.

First, it seems that the complex can bind to hairpins (i.e. ED Fig. 1c, and same for yeast Spo11 complex), so the statement (ED Fig. 6a) that two complexes are bound to the preferred site in the slower migrating band of the EMSA is not fully convincing.

Second, what is the activity of the complex on a similar linear DNA without the preferred cleavage site?

Third, the prediction is that most DSB cleavage on this substrate should be at the inserted cleavage site. One cannot evaluate this with the gel provided (4g) because there is no size marker and the resolution is too low. A higher-resolution gel with a size marker should answer this question.

9) Dimerization: The attempt to enhance cleavage by stabilizing the dimer is interesting. The enhancement is detected but it may be more informative to test various concentrations and kinetics which may reveal an even stronger stimulation. Also, this assay was probably done in reaction conditions with Mn; how is the activity in the presence of Mg? It would be interesting to know if the proportion of topoisomers is reduced with LZ-Spo11, which would argue that these products may

be generated by improper dimerization. The proportion of nicked products does not seem to be reduced from the gel (ED Fig. 6g), or maybe slightly? Also, formally, the authors do not know if the effect of the LZ-Spo11 on cleavage activity is due to the dimerization property of the leucine zipper. Ideally, a LZ zipper deficient for dimerization would address this question.

Discussion: The authors present and discuss how several components may participate to the activity (and assembly) of the active tetrameric complex. One component is the RMM complex, which by forming condensates can provide an organization at loop bases, and the clustering of the Spo11/Top6bl complex and potentially its co-orientation (as discussed in previous papers). Another feature is the direct interaction between Top6bl and Rec114 which along with the structure of the Rec114-Mei4 complex (yeast and mouse) provide a potential framework for the dimerization of the Spo11/Top6bl complex, given the 2:1 stoichiometry, as proposed in the studies by Nore et al. (2022), Daccache et al. (2023), Liu et al. (2023), and Laroussi et al. (2023). These data should thus be integrated in the discussion.

In the supplementary discussion, the authors indicate that whether nicks accumulate in vivo has not been definitively established. This is right, but they were two specific studies that looked for single-strand nicks and failed to detect them at specific loci in yeast (Liu et al., 1995, EMBO J. 14:4599-608; de Massy et al., 1995 EMBO J., 1995, 14:4589-98); it would be worthwhile to mention them.

Minor points:

Legend to Fig. 4: Information about blue and red bars in b and others is missing.

L1009: Recombinant Spo11? I assume the purified Spo11/Top6bl complex?

ED Fig. 3a: Please clarify the interaction seen between the Cter of Top6bl and Spo11: which residues of Spo11 are involved?

ED Fig. 3d: Is there any information that can be gained from the bendability of DNA duplexes and could this be correlated with the cleavage preference?

ED Fig. 7e: It is not easy to track colours. The mouse model should be in gray? And the monomer in pink? The position of the catalytic tyrosine could be added.

Version 2:

Reviewer comments:

Referee #1

(Remarks to the Author)

The authors have done a good job addressing all reviewer concerns, and I support publication.

Referee #2

(Remarks to the Author)

The authors have done a great job of addressing the review concerns, and I fully support publication of this important paper, The new rapamycin-induced dimerization experiment is dope!

Referee #3

(Remarks to the Author)

The authors have resolved all my concerns. A very nice paper.

Referee #4

(Remarks to the Author)

The authors did a great job in the revision and answered to all my comments. This is a great paper and very important findings.

The ligation activity is discussed, and I agree that it is better not to include it in the abstract.

The use of the fusion proteins FRB/FKBP provides more convincing evidence for the role of dimerization on the activity, with the use of rapamycin thus showing directly the impact of dimerization, and thus strengthens the manuscript.

The supplemental discussion is also quite useful to address several interpretations/implications from the data.

Minor comments:

Fig. 4g legend, L552: M-DSB: I assume this is an oligo of 22 bp? If the DSB product comigrates with the marker (or slightly slower on the gel?), why should the cleavage not be at the preferred sequence?

ED Fig. 3b: It is interesting to see that the A TOP6BL interacts with the B SPO11. Also, the conserved motif upstream of the interaction (KAEIQAL in *M. musculus*) corresponds to the conserved motif T2BI defined by Takahashi et al., NAR, 2000 (with potential function discussed in this paper). I also agree that only speculations can be made at this stage.

Typos:

L700: space

L1147: detectable

L1148: non physiological

We thank the reviewers for their overall positive response and constructive suggestions. We have addressed all of the comments through addition of new data, text revisions, and/or rebuttal below. We also substantially rewrote to shorten the text to meet Nature formatting guidelines.

Reviewer comments are in black below, our responses are in blue text. Line numbers refer to the clean Word document without tracked changes.

Referees' comments:

Referee #1:

The manuscript "Reconstitution of SPO11-dependent double-strand break formation" by Zheng...Keeney reports the reconstitution of SPO11-mediated DNA breaks in vitro, an achievement that can reasonably be called a holy grail of the meiosis field, and one that has been 27 years in the making. Because of its obvious importance and broad appeal, plus the fact that the experiments are all of exceptional quality and clearly explained, I strongly support publication after the authors address only a few questions and concerns, detailed below.

Thank you for the positive evaluation.

1. Fig. 1f: Can the authors comment on whether the observed SPO11 complexes in AFM are monomeric (1:1), dimeric (2:2), or larger aggregates, based on the size of the protein blobs bound to DNA?

Unfortunately no, our AFM measurements don't have the sensitivity to evaluate this for protein particles bound to DNA because it is difficult to correct for the contribution of the DNA itself. (This can be done, but it requires more sophisticated methods than we have available to us at present.) We can fairly reliably measure the size of free protein particles. These match well with expectation for monomers (1:1) as we previously showed for yeast, but we decided not to include these data because the mass photometry is a more sensitive and precise way to measure particle sizes.

2. I am as big an AlphaFold fan as you'll find. But, (like the authors I suspect), I do find the lack of two-fold symmetry in AF3 models of DNA-bound TOP6BL-SPO11 complex to be... weird. Usually AlphaFold seems to go out of its way to predict perfectly symmetric complexes. I'm not going to suggest taking this analysis out, but can the authors add a bit more explanation regarding the reproducibility of the exact mode of DNA binding depicted in Fig. 3d across the many models they ran?

Thanks for this question. We too were surprised by AF3 predicting an asymmetric model, but we do not have a lot of experience with this so it is not clear to us how often AF3's models are or are not symmetric. We revisited this systematically for all 25 of the models we present in the paper by compiling the distances between each catalytic tyrosine and its "correct" scissile phosphate (defined relative to the center of the bend). Only one model had nearly identical distances for the two tyrosines, and only another three models had roughly symmetrical distances that were also small ($\leq 3 \text{ \AA}$). The rest of the models had substantial asymmetry or very long Y138-phosphate distances. The model we showed in the paper is in the middle of the pack for both symmetry and distance. Since most of the AF3 models are asymmetric to some degree, we decided to stick with this one as being representative. We added a figure showing

the symmetry measurement (new ED Fig 3h) and added text to explain (Supplemental Discussion 2).

We also revisited the question of which phosphates correspond to the “scissile” ones in the AF3 structure we originally showed. We had tried to force this call to match the position predicted by the sequence preference, but in fact the center of the bend is one base pair over. That is, AF3 did not predict binding at the “correct” site. We have updated the contact map figure (Fig. 3d) accordingly.

3. On the topic of AlphaFold, were all predictions run with preferred DNA sequences according to the preferences described in Fig. 4? If so, what happens with DNA sequences that are designed specifically to be non-preferred?

All of the AF3 models we present have DNA sequences that contain a match to the preferred base composition, but AF3 often placed the bend and the protein active sites somewhere other than the “right” position on the DNA. In fact, even for the same input sequence, models differed with respect to the protein position on the DNA. You can see the signature of this variation in the model colored by pLDDT score (ED Fig. 3a). The DNA has a low score (red color in ED Fig. 3a) even though the DNA bending path is highly superimposable (ED Fig. 3d). The low score arises because, although the bending conformation is reproducible, the predicted bend position is not.

We also ran predictions for another set of DNA sequences not shown in the paper, including ones that do not match the consensus. AF3 generated very similar models with all DNA sequences tested whether they had a good match to the consensus or not. It thus appears that AF3 is not reliably detecting the fine-scale sequence determinants that shape the actual SPO11 cleavage probability. This could mean that AF3 is not yet able to model the effects of DNA sequence on SPO11 binding and DNA bending, or it could mean that the main effects of sequence are on catalysis and not on DNA binding. Since there was no clear specificity in the AF3 models, we elected not to pursue this further in the paper. We added commentary about this to Supplementary Discussion 2.

4. The prevalence of nicks in the biochemical experiments is interesting, and I have a few questions on this topic. First, the prevalence of nicks generated by dimeric complexes in vitro suggests that nicking may occur at a relatively high frequency in vivo. Is there any way (in existing datasets) to support this idea? For future work, would there be any way to detect nicks generated by SPO11 complexes in cells?

When people have looked for nicks in *S. cerevisiae*, they have not seen positive evidence for them, suggesting that they either do not form in appreciable numbers relative to DSBs, or they have a much shorter lifespan than DSBs (Liu et al. PMID: 7556103; De Massy et al. PMID: 7556102). To our knowledge, there is no relevant data about nicking in vivo in mice, and it is not clear to us that available methods would be able to detect them unambiguously.

5. In Fig. 3h, can the authors explain why Unreacted (supercoiled) and Relaxed CCC run differently on a gel with EtBr? Aren't they supposed to be the same molecule (chemically), and doesn't EtBr saturate the DNA enough to positively supercoil even DNA that (as purified from bacteria) is negatively supercoiled?

Where different DNA species will run on the gel depends on the EtBr concentration. This experiment was done at a subsaturating concentration (1 $\mu\text{g/ml}$), so the molecules that start out mostly relaxed become more positively supercoiled than the unreacted substrate, which started out more negatively supercoiled. As a consequence, the relaxed products migrate more rapidly in EtBr than the unreacted substrate does.

6. I would recommend caution in interpreting the finding that the SPO11-TOP6BL complex can mediate topoisomerase-like activity in vitro. As the authors rightly point out, this activity seems limited to the cases where the complex generates a nick, and therefore the dimer is kept close together even while one strand is broken. Given the chemistry of the reaction, it's almost inevitable that relegation can happen in that case (sometimes with relaxation via rotation about the remaining strand, a la Topo I). This is almost certainly not true "type II topoisomerase" strand passage, and as such I would urge restraint when describing. I think the authors did a good job of this, except I would suggest leaving this result out of the abstract.

We agree that this is not a type II topoisomerase activity (or even a type 1A strand passage activity). The mechanism we propose is a swiveling reaction conceptually similar to that of a type 1B enzyme. We added text to **Supplemental Discussion 3** to make this clearer. While we agree that religation might be expected as long as the dimer stays together after strand breakage, we disagree that it was necessarily "inevitable" that this would actually happen. The catalytic components are there, but there was no reason a priori to assume that the 3' OH and tyrosyl phosphodiester (plus all other active site elements) must remain in the right conformation to support catalysis of religation. Thus, we think it is an interesting and important observation that SPO11 can in fact reseal broken strands. However, we take the point that this religation and the accompanying DNA relaxation activity are of uncertain physiological relevance, so we removed mention of the topoisomerase activity from the abstract.

7. Does LZ-SPO11 purify as a dimeric complex?

Apparently yes, based on its elution position in SEC (previously shown in ED Fig. 6e and mentioned in the text (formerly line 384)). However, we have now replaced the LZ-SPO11 experiments in favor of ones using FRB and FKBP fusions, which work better and also provide a cleaner controlled experiment. These fusion proteins purify as monomers, as expected, but can dimerize in the presence of rapamycin.

8. Does the LZ-SPO11 complex mediate topoisomerase-style activity? If the dimer interface is stable, it should not (if the authors' model is correct).

Thank you, this is an interesting point. We agree that stabilizing the dimer interface should interfere with relaxation if our model is correct, but we point out that the actual prediction is that relaxation should be inhibited to some unknown degree, not necessarily completely prevented, because the artificial dimerization is not covalent and can thus dissociate and reassociate multiple times on the time scale of these experiments. It does appear to us that the LZ fusion is less active for relaxation, but it is difficult to say for sure with the data we have available, so we looked at this more closely with the FKBP/FRB fusions. Inclusion of rapamycin did indeed appear to inhibit relaxation as judged by a shift in the topoisomer distribution and retention of more of the negatively supercoiled plasmid compared to no rapamycin (nicking reaction in the presence of Ca²⁺) (**Review Fig. 1**). However, we feel that this needs to be explored more rigorously and quantitatively than is possible in the time we were given for revision, so we opted to leave this out of the manuscript.

[REDACTED]

9. Do the authors anticipate that forcing dimerization of the yeast Spo11 core complex (as in the LZ-SPO11 fusion) would enable it to catalyze DSB formation in vitro?

That is the hope, although we note that a negative result would be uninformative. We look forward to testing this, but this is outside the scope of the current manuscript.

10. For AF3 discussions (and possibly other places) - please clarify if rmsd's are for all atoms or only C-alpha atoms (C-alpha rmsd is typically used for these kinds of comparisons).

Yes, this is correct. We clarified in the text.

Finally - congratulations to the authors on this achievement!

Thank you!

Referee #2:

It has long been acknowledged that the SPO11 protein, related to TopoVI, is the catalytic subunit responsible for DSB formation during meiosis, but for 27 years there had been no demonstration that SPO11 can make DSBs in vitro. With this paper and an accompanying paper from Oger and Claeys Bouuaert, the long wait is over. Using the mouse SPO11-TOP6BL dimer, the authors have detected several activities in addition to the DNA-binding activity already shown for the yeast protein: double-strand cleavage; single-strand nicking; and supercoil relaxation. Based on previous structural data and on structural modeling, the authors predict a pre-cleavage complex that contains two SPO11-TOP6BL dimers bound to bent DNA, with the two active sites formed by contributions by both SPO11 monomers. The activity is very inefficient, which is attributed to a combination of inefficient dimer formation and only a minority of dimers having an active conformation. Support for some of these conclusions come from an inter-subunit complementation experiment and from an artificial tethering experiment. In vitro and in vivo cleavage site sequence preferences are similar, supporting the idea that the activities seen in vitro recapitulate the in vivo activity, but this is the extent of the validation. In concluding, authors emphasize the idea that the monomer-dimer dichotomy is important for the regulation of SPO11 activity, with other proteins of the DSB complex, known to form condensates, providing the environment to concentrate Spo11 and promote formation of active dimers.

This is an important advance in the field, and it almost certainly will be cited in every paper on meiotic recombination for the foreseeable future. It opens the door for further biochemical characterization of SPO11's cleavage activity and its regulation. Because the conclusions rely exclusively on behavior of the complex in vitro, and on in silico modeling of the complex's structure, it will be important in the future to provide in vivo validation of the conclusion and to obtain real structures of the proteins and substrate. Now that the door has been opened, it is likely that many will be rushing through.

We appreciate the positive response.

Comments:

1. It is somewhat disappointing that tethering the two SPO11 subunits resulted in only about a two-fold increase in initial rates, somewhat undercutting suggestions of the importance of the monomer-dimer transition. These experiments were performed at 75 mM SPO11, which is close to the optimal concentration for activity of the untethered complex. Is it possible that, by decreasing the SPO11 concentration and/or increasing the substrate concentration, a greater effect might be observed?

We agree that the result was not especially dramatic, although we do point out that there was no specific prediction ahead of time as to just how much stimulation there should be. Fortunately, however, we have found that using the FRB/FKBP inducible dimerization system works substantially better, giving a much bigger increase in activity with dimer stabilization. We don't know why the FRB/FKBP fusions work better, but speculate that the LZ fusion may constrain SPO11 to suboptimal configurations. Because the new version works better and because it allows us to directly tie the increased activity to dimerization because it is dependent on rapamycin, we decided to replace the LZ-SPO11 experiment entirely.

2. As mentioned above, SPO11 has single-strand nicking, single-strand resealing, and double-strand cutting activity. By comparing in the same experiment the kinetics of nick formation, DSB

formation, and supercoil relaxation, could one come up with a possible model for how the reaction proceeds? At a minimum, it would be useful to provide separate plots for nicking and cutting, rather than just a single line for “substrate utilization” (Figs. 2a,h, 3g, etc.).

We agree this is interesting, but the quantification of plasmid cleavage is too suboptimal to allow this to be done cleanly. Specifically, we can't tell if there is more than one nick, we can't tell if linearized molecules are also nicked, and it is difficult to quantify multiple DSBs. Nicks and DSBs can be quantified better on oligo substrates (for which we do already provide the individual quantification), but there is also the issue of resealing, which complicates a kinetic interpretation of the reaction time course. We don't yet have the ability to generate the kind of quantitative data we need.

3. In Fig. 2d, only linear product is immunoprecipitated, but one would expect both linear and nicked products to be covalently linked to SPO11. Perhaps this is because the starting material contained a vast excess of linear over nicked products. It's important to start with material that has both nicked and linear products in it (perhaps an earlier time point, or a reaction run in Mg⁺⁺, or one run at lower temperature?).

We agree that nicked products of SPO11 cleavage should also have covalent protein attachment. To confirm this, we tested it on the short oligonucleotide substrate with the mix of Y138F and E233A mutant proteins (new ED Fig. 7e).

For the experiment in Fig. 2d, we used a larger amount of SPO11 complexes than in the time course experiments so as to achieve a high degree of cleavage. There is only a little bit of nicked DNA in the input lane, less than was already present in the unreacted substrate, so at least some of the nicked DNA in the input would not be expected to have SPO11 bound to it. We also note that every linear product has two SPO11 molecules bound (one on each end), but a nicked circle only has one SPO11 and thus might be less efficiently recovered in the IP. Nicked circles might also break during the IP or subsequent sample handling, giving linears.

4. Line 315 and following: The logic as to why stronger sequence signatures are seen with more complex substrates could have been presented more clearly. Is it possible that this is just a result of the lower number of unique reads with pUC19?

We can rule out the lower number of unique reads with pUC19 being the cause because there is also a difference between the base compositions with *E. coli* vs. yeast DNA as substrate, which had comparable numbers of uniquely mapped reads.

5. While the in silico structural modeling is voluminous, it is limited to comparisons between mouse and yeast. Could more of value be learned by extending the comparisons to other SPO11s?

We do already provide comparison with SPO11 from other species (ED Fig 8e). It is certainly of interest in the long run to do even more such comparisons, but absent a specific question, we suggest that this is outside the scope of the current study.

Referee #3:

Zheng et al. reconstituted SPO11-TOP6BL complex and characterized its DNA cleavage activity in vitro. As revealed during meiosis, SPO11–TOP6BL complexes cleave DNA to form covalent 5' attachments and this process require SPO11 active site residues, divalent metal ions, and SPO11 dimerization. Additionally, SPO11 also showed topoisomerase activity to relax DNA supercoils and reseal nicked DNA. AlphaFold modeling and deep sequencing of in vitro cleavage products further provided valuable information for SPO11 complex. These results are very interesting. This ms is well written. I'd like it.

Thank you for the positive evaluation.

It is necessary to have a comprehensive discussion of the cleavage differences (target DNA and SPO11 complex activity) observed from in vitro and in vivo experiments. It would be better to compare/discuss the conservation and difference for SPO11 complex in different organisms, such as yeast, human, and mouse (maybe with the help of AlphaFold prediction?).

1. Lines 103-104: Cite the figure (Fig. 1d, 25 and 50 nm), and it would be helpful to show the size markers on the gel in Fig. 1d, and also other gels.

We believe it is clear from context that this sentence refers to the same figures that are cited in the immediately preceding sentence (Fig. 1d and ED Fig. 1c). Fig. 1d is an EMSA. Markers are usually not informative for EMSAs, so they are rarely included (other than an unbound DNA control). We did not include size markers in these experiments, in keeping with standard practice.

2. Lines 116-117: Does this mean the structural difference between yeast and mouse? If so, what difference?

Presumably yes, this reflects some sort of structural difference, but in the absence of a structure of the mouse protein bound to a DNA end, we do not know the answer.

3. Lines 161-162: Negative supercoiled DNA was cut more efficiently; what is the interpretation for this? Actually in yeast, it seems more DSBs and COs form in negative supercoil rich regions. Are they related?

The simple answer is that we don't know why negatively supercoiled performs better in this assay. From the time courses, it appears that a higher fraction of DNA molecules may be bound by cleavage-competent complexes for negatively supercoiled substrates. That inference is based on the initial cleavage phase being faster and consuming a higher fraction of the DNA with the negatively supercoiled substrate. Perhaps it is easier for SPO11 to bend negatively supercoiled DNA correctly, or catalysis is stimulated by negative superhelical tension. Because we have no information yet to make more informed conclusions, we have opted to just present the findings as is without trying to speculate further. The reviewer is correct that there have been studies about the relationship between recombination and supercoiling in yeast, but we do not find the currently available evidence connecting supercoiling to DSB formation to be very persuasive. There is also no information allowing us to tie our findings in any meaningful way to available in vivo data, so we prefer not to speculate about it here.

4. Lines 133-234: Would you please clarify the conformational change?

We added text to try to clarify, but the first part of this paragraph and ED Fig. 5g (and its legend) already describe this conformational change in considerable detail (i.e., the relative motion of the WH and Torpim domains).

5. Lines 343-344: Is the preferred base composition related with the formation of negative supercoils or the organization of chromosome loop/axis?

We are unsure what the reviewer has in mind here. We show directly that the same base composition preference is seen irrespective of substrate supercoiling status. And since we see similar preferences in vitro (where no axis proteins are present) as in vivo, it is not obvious that the loop/axis conformation of chromosomes is relevant to the base composition bias.

6. Fig. 1d,e: Size markers for the gels would be helpful; the number of repeats of experiments and error bars are needed in 1e legends. 1f, what about distance between the two binding sites? 1g, cannot see the DNA alone data, please use another color?

As noted above, size markers are not useful for EMSAs. The experiment in ED Fig 1d was conducted only once, but its repetitive nature (repeated protein titrations on different substrates) provides internal replication. Information about replication was added to the legend of ED Fig. 1d; replication information was already provided for all other experiments. For Fig. 1f, we did not evaluate distances between complexes because these experiments were done at low protein concentration such that most DNA molecules had one or no protein complexes bound. We changed the way the data are plotted in Fig. 1g to make the DNA-alone data clearer.

7. Fig. 2a: Linear indicates a single cleavage site on the plasmid? what are those bands (obviously in WT at higher concentrations) between linear and supercoiled bands? Please clarify. 2d, a little bit confused: 2a showed that most supercoiled plasmids are cut once and thus predominantly linear ones. However, here, 2d showed that supercoiled plasmids are cut more than once and thus multiple discrete bands. if so, what caused the difference?

For Fig. 2a, yes, linear indicates a single DSB (with or without an additional nick(s), which we cannot evaluate). The bands in between are covalently closed topoisomers, as described in detail later in the paper. For Fig. 2d, as described in the figure legend, more protein was used here than in Fig. 2a, so there is more total cleavage.

8. Fig. 4g: Please also show the DNA markers/ sizes for the gel; bottom quantification, not sure Error bars are shown, at least they are invisible in nicks.

We do not routinely include size markers in these experiments because we often need to use all of the available lanes and we already knew from initial experiments the migration positions of the various species (uncut, nicked, DSB). For clarity, we replaced the gel image in Fig. 4g with one containing size standards. Error bars are drawn on all graphs when mentioned in the legend. For Fig. 4g, experiments were highly reproducible, so most of the error bars are smaller than the plotting points. Source data is provided, so readers can evaluate this for themselves.

9. ED Fig. 2e: Negatively supercoiled.

Thanks, changed as suggested.

10. ED Fig. 6a-d: Size markers needed.

As above, no markers are needed for EMSAs (ED Fig. 7a), and we did not run markers in the oligonucleotide cleavage assays because we know the migration positions of substrate and products from prior experiments.

11. The manuscript would be further improved if they can compare/discuss the conservation and difference for SPO11 complex in different organisms, such as yeast, human, and mouse (maybe with the help of AlphaFold prediction?).

We already provide an extensive comparison of mouse and yeast SPO11 (multiple places throughout the manuscript), as well as an analysis of SPO11 from multiple species (ED Fig. 8e). It is certainly likely that there will be further interesting findings about SPO11 structural variation across phyla, but absent a specific question here, we feel that this is outside the scope of the present study.

Referee #4:

This manuscript reports the characterization of the DNA cleavage activity by the mouse Spo11/Top6bl complex.

This is a very important and remarkable achievement in the field of meiosis. Spo11 and then Top6bl were identified several years ago, with substantial evidence showing that Spo11 carries the catalytic activity. However, despite several purification attempts, using proteins from various species, no DNA cleavage was obtained.

Here the authors purify a complex of mouse Spo11 and Top6bl. They show that it forms a heterodimer. They analyse its DNA binding properties. They detect DNA cleavage (nicking and linearization) on plasmid DNA (various topological states, including linear), and crucially this activity is abolished in the Spo11Y138F mutant. They show that Spo11 is covalently attached to the 5' ends. Interestingly, by combining Spo11Y138F and Spo11E224 proteins, they detect nicking activity, which provides support for the active complex being a tetramer. Surprisingly, a DNA relaxation activity is detected. Based on AF3, some unanticipated features of the complex are detected (asymmetry at the dyad axis and DNA unwinding). By nucleotide resolution mapping of the cleavage sites, preference for some nucleotides is detected.

Altogether, this is a great piece of work; the data is very clear and the interpretations are convincing. As noted by the authors, there is one major missing property as compared to the predicted *in vivo* activity: the formation of the heterotetramer. This leaves an unknown, which is that double-strand DNA cleavage is detected without detection of the tetrameric form, and it indicates that a small proportion of the purified protein is active for DNA cleavage. This, however, does not lower the interest and significance of the data presented here. I definitively support the publication: it is a breakthrough in the field.

Thank you for the positive comments. However, we respectfully disagree with the assertion that only a small proportion of the purified protein is active for cleavage. We provide direct evidence that most of the protein is active for cleavage if it can form a dimer on DNA, and the high apparent affinity of DNA binding (~2 nM K_d for duplex binding; sub-nanomolar K_d for end binding) suggests that essentially all of the protein can bind DNA. Our data thus suggest that our preparations are highly active when assessed as the percentage of protein molecules that are capable of strand cleavage.

A few points require revisions at various levels.

The overall presentation is clear. Importantly, the methods are carefully presented, which is key specifically for the purification steps. Several clarifications, however, should be provided with respect to DNA binding, ATP binding and DNA cleavage.

1) Protein purification: The authors should discuss and address the question of whether the tag added (FLAG) may alter or not SPO11 conformation. Based on studies in yeast, it is known that Spo11 is sensitive to tag addition (especially at the C-ter). Since getting *in vivo* validation is not reasonable in the context of this study, one needs to know if a linker was added? Which one? What are the predicted consequences on the structure by modelling?

One correction: to our knowledge, only one C-terminally tagged version of yeast Spo11 has been reported to have compromised activity *in vivo* (HA3His6, generated in our lab). Other C-terminal tags have been reported to be fully functional (Myc, Flag, and Protein A; Klein's,

Ohta's, and our labs). As the reviewer notes, it is not straightforward to test function for the tagged mouse protein *in vivo*. We added the protein sequence of the different constructs as a supplemental file and added text to Methods to acknowledge that we cannot exclude effects of the tags, but we do note that even larger tags than the Flag version (leucine zipper and FKBP/FRB) do not appear to interfere with activity. We also note that the N-terminus of SPO11 is predicted to be unstructured, and inclusion of the N-terminal Flag tag did not affect the AF3 models.

2) DNA binding: Why increased concentration (3.1 nM) restores double-end binding to all oligos is not clear.

The short answer is that we don't know because we don't know the conformations that are possible for end-bound SPO11 complexes. Direct structural information is needed to answer this question, so this is outside the scope of the present study.

With the data provided (EMSA and AFM), at the end it is not clear what is the DNA-end specificity of complex. Does the complex preferentially bind to 5' 2-nt overhangs (as the yeast complex) compared to hairpins?

Yes, taken together, the experiments in Fig. 1 and ED Fig. 1 show clearly that the protein has much higher affinity for a 2-nt 5' overhang than for a hairpin end. In Fig. 1d, the first shift indicates a sub-nM K_d and the second shift indicates that second end binding requires ≥ 25 nM protein. By comparison, ED Fig. 1e shows that both ends of the same length DNA substrate (25 bp) are almost fully saturated at 3.1 nM protein if both ends have the 2-nt overhang, rather than the second end being a hairpin. We don't consider it of primary importance for this paper (which is mostly about DNA cleavage) to focus in detail on end binding specificity, but we look forward to additional biochemical and structural experiments in the future.

It would be interesting to have a long DNA fragment (linearized plasmid) with hairpins at both ends (or relaxed DNA) to investigate duplex binding (by gel shift or AFM) (Yeh et al. (2017) detected discrete DNA-protein complexes with *C. elegans* Spo-11, by gel shift).

We are not sure what the reviewer has in mind here. We already present exactly this kind of analysis through quantification of duplex DNA binding with the double-hairpin substrate used for the cleavage assay (ED Fig. 7a). It would be difficult to interpret binding to a longer substrate since there would be a large number of potential binding states, so we do not think this would add useful information. We also don't consider the Yeh study to be informative in this context, because they reported only extremely low affinity DNA binding ($0.47 \mu\text{M } K_d$, three orders of magnitude weaker than what we observe for the mouse protein). It is not clear if this reflects an exceedingly low intrinsic affinity of the *C. elegans* protein compared to yeast and mouse SPO11, or if instead the purified worm protein was mostly inactive. Additionally, in their EMSAs with 80 bp substrates, the shifted species ran as smears at subsaturating concentrations, so we do not agree that they were able to usefully distinguish between different discrete protein-DNA complexes.

In Fig. 1g, the quantification of bending should be clarified: This reviewer assumes that one-end, three- and four-way are excluded for the quantification and that molecules with internal binding on duplex are selected. Is this correct? Ideally, one would like to know if tetramers, even if low in proportion, could be identified. Potentially by immunogold labelling?

Yes, those other binding classes were excluded (the main text referring to this figure had already said that this analysis was for internal duplex binding). We clarified in the figure legend. In principle, yes, one might be able to distinguish dimers from monomers on the DNA, but not with the methods we have available. Fortunately, we present abundant data from other lines of inquiry that directly observe dimer formation and quantify it, so we opted not to pursue the AFM approaches further at this time.

3) ATP binding: Whether Top6bl binds ATP or not is very important to know. The authors discuss this only in the context of the AF3 model. It would be interesting to know if they have assayed ATP binding activity of the complex even if it is probably not so exciting to present a negative result. In addition, they should at least refer to the two studies addressing this question on the Top6bl subunit alone: Diagouraga et al. (2024) and Chen et al. (2024, NAR). In addition, in Top6bl, it is not only the G1 box which is missing but also the G2 (which includes three conserved glycines) is altered (the first conserved glycine is substituted by a serine). The presentation of results (L212-221) should also refer to previous modelling showing that the ATP lid is missing (Nore et al., 2022), as well as in *A. thaliana* MTopVIBL (Vrylinck et al., 2016).

We agree that it is interesting to know whether TOP6BL binds ATP, but as the reviewer also notes, this has been looked at by others already. No one has yet reported any evidence to suggest that the protein does bind ATP. However, we disagree that we have only addressed this in the context of the AF3 model; perhaps the reviewer has overlooked that we tested by mass photometry for an effect of ATP on dimerization (Fig. 1)? We already cited at that point the Diagouraga paper (line 68). We added citation of the Chen et al. paper (Supplemental Discussion 1). We agree that the G2 box is altered, but this does not contradict what we stated in the paper (we said that the G1 box is missing and the other two G boxes are “degenerate”). We focused on the G1 box because our new findings correct a small issue with previous alignments. We agree that the ATP lid is missing, as previously shown by Nore et al. 2022, so we added mention of this (ED Fig. 4f and its legend).

4) DNA cleavage: In the methods, the strategy used for quantification is described, it is, however, not fully clear; it is mentioned that “when multiple DSBs are clearly present, substrate utilization was quantified as disappearance of the supercoiled band”. In which panels was this applied? Otherwise, it is the “fraction of DNA molecules that had sustained at least one detectable nick or DSB” (why “at least”? How are multiple DSB cleavages quantified here? and it is also written that cleavage is underestimated). It would be interesting to have the quantification of multiple cleavages (by quantifying the smear below the size of the linear product) when these products are prominent (such as in Fig. 2d).

We apologize for the lack of clarity. The alternative quantification methods applied to all experiments, as it was a lane-by-lane decision. We now provide details in the Source Data about which method was used. We also provided more information in the Methods to explain the limitations of the quantification of the plasmid cleavage assay.

We face unfortunate limitations to the quantification because we can only see the first cut on a DNA molecule. Once a molecule is nicked, we don't know if it has additional nicks or not. And once a DSB is formed, we don't know if there are also nicks present. We are not attempting to quantify multiple cleavages because we cannot do so unambiguously. Importantly, this has no bearing on any of the conclusions that we draw because we have been careful to keep within the bounds of what we can actually measure. We agree that it would be interesting to be able to quantify multiple cleavages on a single DNA, but these experiments are not the right way to approach this. Quantifying the smear is not a rigorous approach to measuring multiple cutting.

One question that also arises is whether the kinetics of nicked and linear products are identical. Thus, each product should be quantified separately.

We agree that this is an interesting question, but our experiments with plasmid substrates are not appropriate to address it because of the ambiguity in detection as noted in the preceding response. But we do already address the specific question the reviewer asks: are the kinetics for nicked and linear products identical? We explicitly mentioned already that they are not (nicked products predominate at early time points). By contrast, the experiments with the oligo substrate are much more appropriate for this analysis because we can separately quantify nicks and DSBs, and we provide this quantification already. This also clearly demonstrated that nicks form more rapidly. Further experimentation is needed to tease out the relationship between nicking and double-strand cleavage, but that is beyond the scope of this initial study.

In the text, the results of panel 2d cannot be presented without explaining or describing the smear (multiple cuts) in the WT IP lane.

We added explanation to the figure legend.

Overall, this reviewer understands that both nicks and DSBs most likely result from the complex in a tetrameric form but that nicks are abundant due to some instability or improper assembly of the tetramer (improper dimerization of Spo11). If this is correct, this should be more explicitly stated in the description of the results.

There are currently many questions about the relationship between nicks and DSBs that will be interesting to address now that we have an experimentally tractable system. We added a brief discussion of this issue (Supplemental Discussion 5). We don't agree that instability or improper assembly of the dimer is the only possible explanation for nicks (e.g., it could be that some sequences are easily cut on only one of the two strands). In fact, the observation that the large majority of dimer-bound oligonucleotide substrates acquire a DSB rather than a nick (Fig. 4g) argues against there being a substantial population of improperly assembled dimers.

5) Complex assembly: Since most protein is assembled as monomer of Spo11/Top6bl, it is important to understand the dynamic of the activity. The experiment in Fig. 2h is not fully convincing: how do the authors exclude that the cleavage on the plasmid added second may be due to some free Spo11/Top6bl complex rather than resulting from slow exchange?

The reviewer's alternative hypothesis here is unclear. Perhaps the reviewer is positing that there is a subpopulation of free SPO11 at the beginning of the incubation and that that is the only SPO11 available to bind and cleave the substrate added second? Such a model strikes us as implausible since the apparent affinity (~ 2 nM K_d) predicts a binding half-life that is short relative to the length of the time course, and there would thus need to be some unspecified mechanism to constrain SPO11 to rebind only to the original DNA molecule once it dissociates. This would be unusual behavior for any protein, let alone a topoisomerase relative. Moreover, this model would need to explain why there is enough free SPO11 available to cut nearly all of the large plasmid by 30 min (Fig. 2h.ii), but it does so only very slowly (much more slowly than free SPO11 binds during the initial incubation on ice). The model we propose is by contrast very simple and explains these properties of the reaction easily without invoking unusual protein behaviors.

The molar ratio complex/DNA is very high; do the authors imply that all or most of these complexes are bound from the start of the reaction?

Yes, that is exactly what we envision, but the reviewer's statement about the molar ratio emphasizes the wrong point. The protein:DNA molar ratio is high only if one focuses on entire DNA molecules (~40:1 for pUC19, ~100:1 for the larger plasmid). That is not the relevant ratio, however, because each DNA molecule has thousands of potential binding sites and could theoretically accommodate hundreds of SPO11–TOP6BL complexes if they were packed side by side. So, available DNA binding sites are in considerable molar excess over protein in this experiment, leading to the expectation that there will be little free SPO11 protein at equilibrium.

Whether the slope of the cleavage of the second plasmid is the same as the second phase of the first plasmid is not shown by the graphs presented. In the reaction there is probably a mix of monomers (in excess), improperly assembled dimers (may be leading to nicking), and properly assembled dimers (DSB cleavage), and this, in a dynamic status. Please clarify.

We did not claim that the slope of cleavage for the second plasmid is the "same" as the second phase of the first plasmid. Instead, we stated that it is "comparable to the slow portion of the respective two-phase reaction" (meaning the slow phase on the same plasmid in a different mixing regime), and that it appears monophasic rather than biphasic. Because of the substantial limitations that preclude us from accurately quantifying total cleavage in these plasmid reactions (elaborated on in our response to this reviewer's point #4), we are unable to provide a more precise determination of absolute cleavage rates. However, what we do show is sufficient to make the point we are trying to make in this experiment, particularly given the reciprocal behavior of the two mixing regimes, the shapes of the time course curves (biphasic for first plasmid vs. monophasic for second plasmid), and the clear distinction between the sequential mixing regimes and the simultaneous mixing regime.

6) The nicking activity of the complex: The combination of Y138F and E224A is a great experiment showing the role of the expected Spo11 dimer. It is the only experiment allowing to conclude as written on L400, that Spo11 can dimerize on DNA. Evidence for topoisomerase activity with the WT protein should also be validated by EtBr addition (as shown for the mutants in 3h). This unexpected property could be discussed, and specifically, whether it is an in vitro artefact or whether it may have significance in vivo.

We thank the reviewer for the positive comment about the mutant heterodimer experiment, but we disagree that this was the only experiment evidencing the expected dimerization. The experiment showing stimulation by artificial dimerization is also evidence for this.

For the topoisomerase activity: There were already multiple lines of evidence that clearly established that the wild type protein can do this, namely, the prominent ladder (identical in appearance to the mutant heterodimer topoisomer ladder) in the presence of Ca^{2+} despite there being little or no detectable linearization; and the weak ladder in Mn^{2+} without evidence of a topoisomer-matching banding pattern when DNA was more heavily digested (Fig. 2d). Nevertheless, we performed the requested EtBr experiment with wild-type protein and got the expected result (new ED Fig. 4i). We had already explained in Supplemental Discussion that relaxation activity may be disfavored in vivo (see Supplemental Discussion 3, para. 3).

7) DNA sequence preference: The mapping of cleavage on pUC19 includes both single-strand nicks and double-strand cuts. It would be interesting to determine if there is any difference. It is surprising that the authors have used as genomic DNA *E. coli* and yeast DNA. It would be more

logical to test mouse genomic DNA (and to use another DNA as carrier if needed for the library preparation). Is there any specific rationale for this choice?

We didn't use S1 nuclease, so we probably do not recover nicks efficiently, if at all. If nicks were being converted to DSBs by shearing of the intact strand, we could pick it up but it would have a different spatial signal (-1 offset in the cross correlation between strands) that we do not observe in the data. We agree that it will be interesting to look at this in more detail to determine if there are sequence determinants that govern the relative proportions of nicks and DSBs, but this is beyond the scope of the current study.

Our purpose in using yeast and bacterial DNA was simply to expand the repertoire of sequences available for SPO11 to choose among. Mouse genomic DNA would not be any more relevant for this experiment because the in vitro experiments lack the very large array of other components that are well known to play major roles in shaping the DSB landscape in vivo (nucleosomes and histone modifications; PRDM9; higher order chromosome structure; REC114, MEI4 and other DSB proteins; etc.). Since we are not attempting to reconstitute the full in vivo specificity but are instead trying to decipher the intrinsic sequence preferences of SPO11, DNA from any source is equally appropriate. Also, on a practical note, we considered that mouse genomic DNA might be too complex to get meaningful results because of the limit to the amount of protein we can use in the in vitro digestion.

8) The hairpin linear substrate: This is an important experiment but somewhat confusing, and several additional information are needed for interpreting this data.

First, it seems that the complex can bind to hairpins (i.e. ED Fig. 1c, and same for yeast Spo11 complex), so the statement (ED Fig. 6a) that two complexes are bound to the preferred site in the slower migrating band of the EMSA is not fully convincing.

The reviewer seems to be objecting to something we had been careful not to say. As noted in response to this reviewer's point #4, our data clearly show that hairpins are poor binding substrates (about two orders of magnitude higher K_d than binding to 5'-overhang ends, and at least an order of magnitude higher than binding internally to a duplex). Note that we explicitly said that the hairpins reduce end binding to this substrate, we did not claim that there is no binding. Also note that we worded things carefully to say that the substrate was designed to "encourage" binding of a single dimeric complex near the middle. We deliberately did not make the claim that this is the only binding mode; the cartoon in ED Fig. 6a was meant to be illustrative of the likely preferred conformation, not to exclude the possibility of other conformations. We clarified in the figure legend.

Second, what is the activity of the complex on a similar linear DNA without the preferred cleavage site?

We have not yet tested this. However, we do want to emphasize that our data showing that SPO11 prefers a particular base composition does NOT demonstrate (and we do not claim) that that is the only sequence it is capable of binding and cleaving. Indeed, we show the opposite: given a low-complexity DNA substrate, SPO11 will readily make use of sequences that it prefers not to use when in the context of a high-complexity substrate. We agree that it will be interesting to explore the sequence preferences in more detail with the oligonucleotide substrates, but this will be a substantial undertaking to carry out rigorously.

Third, the prediction is that most DSB cleavage on this substrate should be at the inserted cleavage site. One cannot evaluate this with the gel provided (4g) because there is no size

marker and the resolution is too low. A higher-resolution gel with a size marker should answer this question.

The reviewer is correct that our expectation in designing the substrate was that SPO11 would use the same site that it uses when in pUC19. However, this is not really an essential prediction because of the aforementioned plasticity of SPO11 site usage, particularly when presented with a low-complexity target. In the cleavage reactions with pUC19 or yeast or *E. coli* genomic DNA, the DNA is in considerable excess over protein. In the oligonucleotide cleavage experiments, in contrast, protein is in substantial excess such that a large fraction of the low-complexity substrate is bound by two SPO11–TOP6BL complexes. It is therefore possible that SPO11 is able to use a different cleavage site under these conditions than it would prefer to have used if the same sequence was in the full pUC19 context. We have not mapped the cleavage position precisely, but this information is not needed to interpret the experiments for the purpose that they serve in this paper. We replaced the gel image in Fig. 4g with one containing size standards, and provided further information in the figure legend.

9) Dimerization: The attempt to enhance cleavage by stabilizing the dimer is interesting. The enhancement is detected but it may be more informative to test various concentrations and kinetics which may reveal an even stronger stimulation. Also, this assay was probably done in reaction conditions with Mn; how is the activity in the presence of Mg? It would be interesting to know if the proportion of topoisomers is reduced with LZ-Spo11, which would argue that these products may be generated by improper dimerization. The proportion of nicked products does not seem to be reduced from the gel (ED Fig. 6g), or maybe slightly? Also, formally, the authors do not know if the effect of the LZ-Spo11 on cleavage activity is due to the dimerization property of the leucine zipper. Ideally, a LZ zipper deficient for dimerization would address this question.

To formally prove that the stimulation is related to dimerization per se, we replaced the LZ-SPO11 experiments with the FKBP/FRB version. We agree that some of these other questions are interesting to address, but we suggest that they are outside the scope of the current study. As above, we do not agree with the reviewer's assertion that nicks must arise from improper dimerization, so we do not agree that there is a strong prediction about what should happen vis a vis nicking activity in these experiments. For example, one could envision that enhanced dimer formation stimulates DSB formation in some contexts (as the reviewer argues) but actually enhances nicking in other contexts (e.g., by stabilizing binding and thereby favoring catalysis on sites that are less optimal for double-strand breakage), such that the population of products shifts in unpredictable ways. Again, this is interesting to consider and to address, but it is actually quite difficult to address rigorously with complex substrates.

Discussion: The authors present and discuss how several components may participate to the activity (and assembly) of the active tetrameric complex. One component is the RMM complex, which by forming condensates can provide an organization at loop bases, and the clustering of the Spo11/Top6bl complex and potentially its co-orientation (as discussed in previous papers). Another feature is the direct interaction between Top6bl and Rec114 which along with the structure of the Rec114-Mei4 complex (yeast and mouse) provide a potential framework for the dimerization of the Spo11/Top6bl complex, given the 2:1 stoichiometry, as proposed in the studies by Nore et al. (2022), Daccache et al. (2023), Liu et al. (2023), and Laroussi et al. (2023). These data should thus be integrated in the discussion.

We had previously considered and discussed amongst ourselves the possibility that the 2:1 stoichiometry of the REC114-MEI4 complex plus the interaction between TOP6BL and REC114 might support dimerization. However, we do not agree amongst ourselves that this is a

compelling model because there are multiple lines of evidence indicating that the functional REC114-MEI4 entity in vivo is a higher order assembly rather than an isolated REC114-MEI4 heterotrimer. Given what is currently known about the ability of yeast RMM condensates to recruit Spo11 complexes, we do not see any reason to favor the idea that the two SPO11 complexes that make up a catalytically active dimer must be both bound to the same REC114-MEI4 heterotrimer. Nonetheless, we added some text to address this in Supplemental Discussion 6.

In the supplementary discussion, the authors indicate that whether nicks accumulate in vivo has not been definitively established. This is right, but they were two specific studies that looked for single-strand nicks and failed to detect them at specific loci in yeast (Liu et al., 1995, EMBO J. 14:4599-608; de Massy et al., 1995 EMBO J., 1995, 14:4589-98); it would be worthwhile to mention them.

Thank you, citations added

Minor points:

Legend to Fig. 4: Information about blue and red bars in b and others is missing.

Thank you, keys added.

L1009: Recombinant Spo11? I assume the purified Spo11/Top6bl complex?

Yes, thanks. Corrected (now line 891)

ED Fig. 3a: Please clarify the interaction seen between the Cter of Top6bl and Spo11: which residues of Spo11 are involved?

We added a figure (new ED Fig. 3b) to show the details of the contacts, and added text to further describe (Supplemental Discussion 2).

ED Fig. 3d: Is there any information that can be gained from the bendability of DNA duplexes and could this be correlated with the cleavage preference?

We have looked into this but have found no obvious connection to the intrinsic bendability of DNA. The relevant issue may be SPO11 protein-induced bendability, but we are not currently in a position to explore this rigorously.

ED Fig. 7e: It is not easy to track colours. The mouse model should be in gray? And the monomer in pink? The position of the catalytic tyrosine could be added.

(Now ED Fig. 8e) We added labels to further clarify and indicated the position of the catalytic tyrosines.

Reviewer comments are in black text, responses in blue.

Referee #1:

The authors have done a good job addressing all reviewer concerns, and I support publication.

Thank you

Referee #2:

The authors have done a great job of addressing the review concerns, and I fully support publication of this important paper, The new rapamycin-induced dimerization experiment is dope!

Thank you! We are really pleased with that experiment too.

Referee #3:

The authors have resolved all my concerns. A very nice paper.

Thank you

Referee #4:

The authors did a great job in the revision and answered to all my comments. This is a great paper and very important findings.

The ligation activity is discussed, and I agree that it is better not to include it in the abstract.

The use of the fusion proteins FRB/FKBP provides more convincing evidence for the role of dimerization on the activity, with the use of rapamycin thus showing directly the impact of dimerization, and thus strengthens the manuscript.

The supplemental discussion is also quite useful to address several interpretations/implications from the data.

Thank you for the positive comments

Minor comments:

Fig. 4g legend, L552: M-DSB: I assume this is an oligo of 22 bp? If the DSB product comigrates with the marker (or slightly slower on the gel?), why should the cleavage not be at the preferred sequence?

The nicked product migrates more slowly than the marker, most likely because of residual amino acid(s) not removed by proteinase K. We would therefore expect the DSB product to also migrate more slowly than the marker for the same reason, but it more closely comigrates with the marker. We therefore cannot exclude the possibility that the cleavage is happening somewhere other than the intended site.

ED Fig. 3b: It is interesting to see that the A TOP6BL interacts with the B SPO11. Also, the conserved motif upstream of the interaction (KAEIQAL in *M. musculus*) corresponds to the conserved motif T2BI defined by Takahashi et al., NAR, 2000 (with potential function discussed in this paper). I also agree that only speculations can be made at this stage.

Thank you

Typos:

L700: space

Thank you, corrected

L1147: detectable

Thank you, corrected

L1148: non physiological

Nonphysiological (original text) is correct